# Molecularly defined and spatially resolved cell atlas of the whole mouse brain

Meng Zhang[1,2,3,5], Xingjie Pan[1,2,3,5], Won Jung[1,2,3,5], Aaron R. Halpern[1,2,3], Stephen W. Eichhorn[1,2,3], Zhiyun Lei[1,2,3], Limor Cohen[1,2,3], Kimberly A. Smith[4], Bosiljka Tasic[4], Zizhen Yao[4], Hongkui Zeng[4] & Xiaowei Zhuang[1,2,3 ✉]

In mammalian brains, millions to billions of cells form complex interaction networks to enable a wide range of functions. The enormous diversity and intricate organization of cells have impeded our understanding of the molecular and cellular basis of brain function. Recent advances in spatially resolved single-cell transcriptomics have enabled systematic mapping of the spatial organization of molecularly defined cell types in complex tissues[1–3], including several brain regions (for example, refs. 1–11). However, a comprehensive cell atlas of the whole brain is still missing. Here we imaged a panel of more than 1,100 genes in approximately 10 million cells across the entire adult mouse brains using multiplexed error-robust fluorescence in situ hybridization[12] and performed spatially resolved, single-cell expression profiling at the whole-transcriptome scale by integrating multiplexed error-robust fluorescence in situ hybridization and single-cell RNA sequencing data. Using this approach, we generated a comprehensive cell atlas of more than 5,000 transcriptionally distinct cell clusters, belonging to more than 300 major cell types, in the whole mouse brain with high molecular and spatial resolution. Registration of this atlas to the mouse brain common coordinate framework allowed systematic quantifications of the cell-type composition and organization in individual brain regions. We further identified spatial modules characterized by distinct cell-type compositions and spatial gradients featuring gradual changes of cells. Finally, this high-resolution spatial map of cells, each with a transcriptome-wide expression profile, allowed us to infer cell-type-specific interactions between hundreds of cell-type pairs and predict molecular (ligand–receptor) basis and functional implications of these cell–cell interactions. These results provide rich insights into the molecular and cellular architecture of the brain and a foundation for functional investigations of neural circuits and their dysfunction in health and disease.

Mammalian brain functions are orchestrated by coordinated actions and interactions of numerous different cell types. Single-cell RNA sequencing (scRNA-seq) provides a systematic approach to classify cell types through gene expression profiling of individual cells[13]. Single-cell epigenomic profiling further enables systematic characterizations of gene-regulatory signatures of different cell types[14,15]. Numerous molecularly distinct cell types have been identified in the brain using these methods. For example, several hundred transcriptionally distinct cell populations have been identified across the entire mouse brain through scRNA-seq of approximately 500,000–700,000 cells[16,17]. However, the limited sampling sizes in these studies have probably led to an underestimation of the cellular diversity of the brain. Moreover, understanding the molecular and cellular mechanisms underlying brain functions requires not only a comprehensive classification of

cells and their molecular signatures, but also a detailed characterization of the spatial organization and interactions of molecular defined cell types. For example, the layered organization of the cerebral cortex and the nucleus organization in the hypothalamus directly impact their functions. At a finer scale, spatial relationship between cells is a major determinant of cell–cell interactions and communications through juxtacrine and paracrine signalling. While synaptic communications can occur between neurons whose cell bodies are far apart, interactions between neurons and non-neuronal cells, as well as interactions among non-neuronal cells, often occur through direct soma contact or paracrine signalling and hence require spatial proximity between cells. In addition, interactions involving local interneurons also tend to occur between spatially proximal neurons. Therefore, a high-resolution, spatially resolved cell atlas of the whole brain would

[1]Howard Hughes Medical Institute, Harvard University, Cambridge, MA, USA. [2]Department of Chemistry and Chemical Biology, Harvard University, Cambridge, MA, USA. [3]Department of Physics, Harvard University, Cambridge, MA, USA. [4]Allen Institute for Brain Science, Seattle, WA, USA. [5]These authors contributed equally: Meng Zhang, Xingjie Pan, Won Jung. ✉e-mail: zhuang@chemistry.harvard.edu

provide an invaluable resource for understanding how the brain functions.

## MERFISH imaging of the whole mouse brain

Recent advances in spatially resolved transcriptomics have enabled gene-expression profiling and cell-type identification while maintaining the spatial information of cells in intact tissues[1,2]. Here we used multiplexed error-robust fluorescence in situ hybridization (MERFISH)[12], a spatially resolved single-cell transcriptomics method, to generate a comprehensive, molecularly defined and spatially resolved cell atlas of the entire adult mouse brain. We selected a panel of more than 1,100 genes (Supplementary Table 1) for MERFISH imaging. These genes were selected based on a whole-brain scRNA-seq dataset described in a companion article in this BICCN package[18]. Analysis of the scRNA-seq data resulted in 5,322 cell clusters, which were grouped into 338 subclasses[18], and our MERFISH gene panel was selected from marker genes differentially expressed among these cell populations (Methods and Fig. 1a).

We imaged these genes in 245 total coronal and sagittal sections spanning whole hemispheres of four adult mouse brains, including serial coronal sections at 100-μm intervals (animal 1, female) or 200-μm intervals (animal 2, male), and serial sagittal sections at 200-μm intervals (animals 3 and 4, male) (Methods; Fig. 1a). Individual RNA molecules were identified and assigned to cells segmented based on DAPI and total RNA signals, providing the expression profiles of individual cells (Methods). Our MERFISH data exhibited excellent reproducibility between replicate animals (Extended Data Fig. 1a). The mean copy number per cell for individual genes obtained from MERFISH correlated well with whole-brain bulk RNA-seq and scRNA-seq data (Extended Data Fig. 1b,c). In total, we imaged and segmented approximately 10 million cells across the adult mouse brain, including all 11 major brain regions: olfactory areas, isocortex (CTX), hippocampal formation, cortical subplate, striatum, pallidum, thalamus, hypothalamus, midbrain, hindbrain and cerebellum. Among the approximately 10 million cells, 9.3 million passed the cell volume and doublets quality controls (Methods).

## Cell classification and registration to the CCF

We integrated MERFISH data with scRNA-seq data using a canonical correlation analysis-based method[19] and classified the MERFISH cells using k-nearest neighbour classification (Methods; Fig. 1a). These two datasets integrated well with each other (Fig. 1b, left, and Extended Data Fig. 1d), and the cell-type labels were transferred from the scRNA-seq cells to the MERFISH cells with high-confidence scores (Methods; Extended Data Fig. 1e). We set a threshold on the confidence scores for cell-type label transfer (0.8 for subclass label transfer and 0.5 for cluster label transfer; see Methods). Among the MERFISH cells, 83% and 74% passed the subclass and cluster confidence score thresholds, respectively, and were used for subsequent analysis. We further validated the robustness of label transfer by classifying MERFISH cells using an alternative method based on transcriptional similarity of MERFISH cells to the mean expression profiles of scRNA-seq clusters. Results from these two methods showed excellent agreement (Extended Data Fig. 1f). All 338 subclasses and more than 99% (5,275) of the 5,322 clusters identified by scRNA-seq were observed in the MERFISH data with the set label-transfer confidence score thresholds. Integration of MERFISH and scRNA-seq data also allowed us to impute the transcriptome-wide expression profile for the MERFISH cells (Methods), which showed excellent agreement with direct MERFISH measurements and the Allen Brain Atlas in situ hybridization data[20] (Methods and Extended Data Fig. 2). To enable systematic quantifications of the cell-type composition and organization in different brain regions, we registered the cell atlas generated by MERFISH to the Allen Mouse Brain Common Coordinate Framework version 3 (CCFv3)[21] using both the DAPI images

and the cell-type-based landmarks (Methods, Fig. 1a and Extended Data Fig. 3). This CCF registration allowed us to place each individual MERFISH-imaged cell, with its cell-type-identity label, into the 3D CCF space (Fig. 1b (right), 1c, 1d and Extended Data Fig. 3b).

The spatial information measured by MERFISH was also used for the annotations of the cell types identified by scRNA-seq, as described in the companion paper[18]. In brief, except for some previously well-annotated cell types, each neuronal subclass name has three parts: the brain region where the subclass primarily resides, one or more marker genes, and the major neurotransmitter used. Non-neuronal cell subclasses were annotated based on marker genes and named based on previous knowledge (microglia, astrocyte, among others) with spatial information being specified in some cases. The cell clusters were named by the subclass names followed by numerical indices in most cases.

## Diversity and spatial organization of neurons

Registration of the MERFISH-derived cell atlas to the Allen CCF allowed us to quantify the composition of cell types in individual brain regions (Fig. 1d). Overall, the whole mouse brain consisted of 46% neurons and 54% non-neuronal cells. This ratio varied substantially from region to region, with the hindbrain and cerebellum showing the lowest and highest neuronal-to-non-neuronal cell ratio, respectively (Fig. 2a).

Neurons exhibited an exceptionally high level of diversity, comprising 315 subclasses and more than 5,000 clusters (see Supplementary Table 2 for the neuronal cell-type composition in the 11 major brain regions). Neuronal cell types also exhibited strong regional specificity with most neuronal subclasses being only enriched in one of the 11 major regions and some spanning multiple, usually physically connected, regions (Fig. 2b). Many of the subclass boundaries aligned well with the region boundaries in the CCF. For example, the intratelencephalic (IT) subclasses showed a clean separation at the boundaries between the isocortex and olfactory areas or hippocampal formation (Extended Data Fig. 4a). In the thalamus, AV Col27a1 Glut and AD SerpinB7 Glut perfectly fit in the anteroventral and anterodorsal nucleus, respectively (Extended Data Fig. 4b). Some subclasses spanned multiple brain regions. For example, inhibitory neuronal subclasses marked by *Lamp5*, *Sncg*, *Vip*, *Sst* or *Pvalb* were distributed across the isocortex, hippocampal formation, olfactory areas and cortical subplate (Extended Data Fig. 4c), consistent with previous findings[22,23].

The 11 major regions contained different numbers of cell types (Supplementary Table 2). In particular, the hindbrain, midbrain and hypothalamus contained substantially greater number of neuronal cell types than the other brain regions (Fig. 2b). We further quantified the local complexity of neuronal cell-type composition, defined as the number of distinct neuronal subclasses present in the 50 nearest spatial neighbours of each cell. Of note, the local complexity was also substantially higher in the midbrain, hindbrain and hypothalamus (Extended Data Fig. 4d), indicating that these regions were not simply composed of more subregions with distinct cell compositions, but also complex local neighbourhood with higher cellular diversity. In addition, some other brain regions also contain small subregions with high local cell-type composition complexity (Extended Data Fig. 4d).

## Spatially dependent neurotransmitter and neuropeptide usage of neurons

On the basis of the expression of neurotransmitter transporters and genes involved in neurotransmitter biosynthesis, we classified matured neurons into eight partially overlapping groups: glutamatergic (expressing *Slc17a7*, *Slc17a6* and/or *Slc17a8*), GABAergic (expressing *Slc32a1*), serotonergic (expressing *Slc6a4*), dopaminergic (expressing *Slc6a3*), cholinergic (expressing *Slc18a3*), glycinergic (expressing *Slc6a5*), noradrenergic (expressing *Slc6a2*) and histaminergic (expressing *Hdc*) neurons.

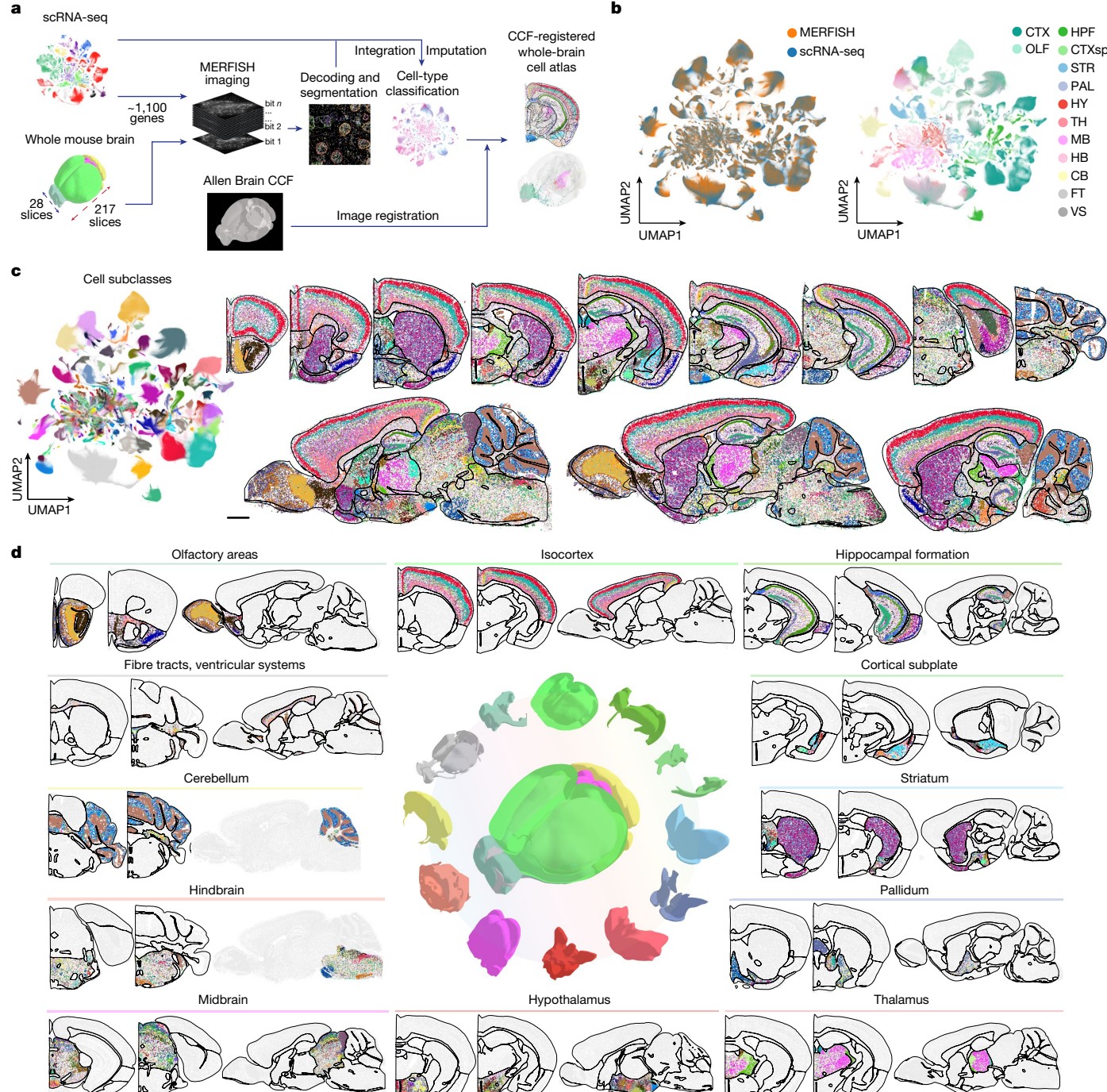

**Fig. 1 | A molecularly defined and spatially resolved cell atlas of the whole mouse brain. a**, Workflow to construct a whole mouse brain cell atlas. A panel of genes were chosen for MERFISH imaging based on the clustering results from scRNA-seq data. MERFISH images were decoded and segmented, and the resulting single-cell gene expression profiles were integrated with scRNA-seq data to classify MERFISH cells and impute transcriptome-wide expression profiles. Finally, MERFISH images were registered to the Allen CCFv3 (ref. 21). **b**, Uniform manifold approximation and projection (UMAP) of the integrated scRNA-seq and MERFISH data with cells coloured by experimental modalities (left) or by major brain regions in which the registered cells reside (right). The number of cells in the MERFISH or scRNA-seq dataset in each subclass was downsampled to the corresponding number in the other dataset for visualization purpose. The UMAP with all MERFISH and scRNA-seq cells displayed is shown in Extended Data Fig. 1d. CB, cerebellum; CTX, isocortex; CTXsp, cortical subplate; FT, fibre tract; HB, hindbrain; HPF, hippocampal formation; HY, hypothalamus;

MB, midbrain; OLF, olfactory area; PAL, pallidum; STR, striatum; TH, thalamus; VS, ventricular system. **c**, UMAP of the integrated MERFISH and scRNA-seq data (left). Spatial maps of the cell types in example coronal and sagittal sections are also shown (right). Cells are coloured by their subclass identities. The black lines in the brain spatial maps here and in subsequent figures mark the major brain region boundaries defined in the CCF[21]. Scale bar, 1 mm. In this and subsequent figures, all cells are shown in the experimental coordinates and the boundaries of brain regions were transformed to the experimental coordinates based on our CCF registration results (Methods). **d**, Spatial maps of example coronal and sagittal sections in the 11 major brain regions as well as in fibre tracts and ventricular systems. Cells are coloured by their subclass identities as in **c**. The underlying contour lines marking brain region boundaries in **a**, **c** and **d** and the 3D brain contours in **a** and **d** were generated using coordinates from the Allen Mouse Brain CCFv3 (ref. 21).

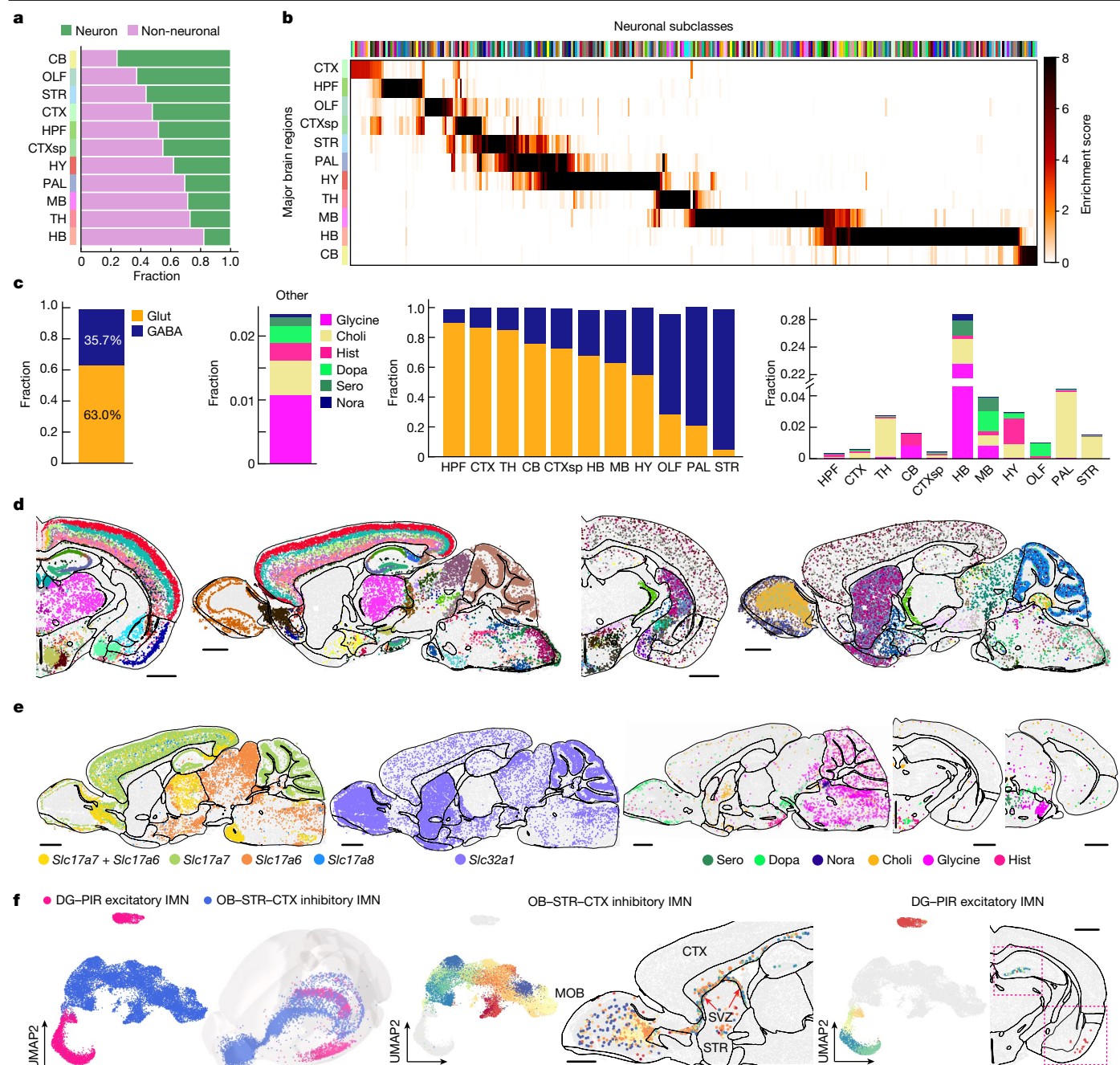

**Fig. 2 | Cell-type compositions and spatial distributions of neurons.**
**a**, Fractions of neurons and non-neuronal cells in the 11 major brain regions.
**b**, Heatmap showing the enrichment score of each neuronal subclass in the 11 major brain regions. The enrichment score is defined as the fold change of the average cell density of a subclass within a brain region compared with the average density across the whole brain. **c**, Bar plots showing the fractions of neurons using different neurotransmitters across the whole brain (left two panels) and in individual brain regions (right two panels). Choli, cholinergic neuron; dopa, dopaminergic neuron; GABA, GABAergic neuron; glut, glutamatergic neuron; glycine, glycinergic neuron; hist, histaminergic neuron; nora, noradrenergic neuron; sero, serotonergic neuron. **d**, Spatial maps of the glutamatergic (left) and GABAergic (right) neuronal subclasses in example coronal and sagittal sections, with cells coloured by their subclass identities. **e**, Spatial maps of glutamatergic neurons expressing *Slc17a7*, *Slc17a6*, *Slc17a7* + *Slc17a6* and

*Slc17a8* (left), GABAergic neurons (middle) and neurons expressing various modulatory neurotransmitters (right). **f**, UMAP and spatial distribution of the immature neurons (IMNs) shown in the 3D CCF space, with cells coloured by subclass identities (left). UMAP and spatial distribution of the inhibitory IMNs shown in a sagittal section, with cells coloured by cluster identities (middle). Excitatory IMNs are shown in grey in the UMAP. UMAP and spatial distribution of the excitatory IMNs shown in a coronal section, with cells coloured by cluster identities (right). Inhibitory IMNs are shown in grey in the UMAP. Scale bars, 1 mm (**d**–**f**). DG, dentate gyrus; MOB, main olfactory bulb; PIR, piriform area; SVZ, subventricular zone. The red boxes mark the two locations of the excitatory IMNs in DG and PIR, respectively. The underlying contour lines marking brain region boundaries in **d**–**f** and the 3D brain contours in **f** were generated using coordinates from the Allen Mouse Brain CCFv3 (ref. 21).

Among these groups, glutamatergic and GABAergic neurons accounted for approximately 63% and 36% of the total neuronal populations, respectively, whereas serotonergic, dopaminergic, cholinergic, glycinergic, noradrenergic and histaminergic neurons (often co-expressing glutamate or GABA transporters) accounted for only approximately 2% of the total neuronal population (Fig. 2c, left).

Both glutamatergic and GABAergic neurons were widely distributed across the whole brain and were classified into diverse cell types with distinct spatial distributions (Fig. 2d,e). The glutamatergic-to-GABAergic neuron ratio varied drastically across brain regions (Fig. 2c, middle). The hippocampal formation, isocortex and thalamus had the highest glutamatergic-to-GABAergic neuron ratios, whereas the striatum and pallidum had the lowest. Although the thalamus was mostly made of glutamatergic neurons, the reticular nucleus of the thalamus was exclusively GABAergic (Extended Data Fig. 4e). In the midbrain and hindbrain, glutamatergic and GABAergic neurons were widely distributed in a partially intermingled manner (Extended Data Fig. 4f). In the cerebellum, glutamatergic and GABAergic neurons were separately enriched in the granular and molecular layers, respectively (Extended Data Fig. 4g). A small fraction of neurons (approximately 1%) co-expressed both glutamate and GABA neurotransmitter transporter genes (*Slc17a6/7/8* and *Slc32a1*, respectively) and these neurons were enriched in various non-telencephalic areas such as the globus pallidus internal segment, hypothalamic nuclei such as the anterior hypothalamic nucleus and supramammillary nucleus, and some subregions in the midbrain and hindbrain, as well as in the main olfactory bulb (Extended Data Fig. 4h), corroborating and expanding previous work[4,24–27].

Among the glutamatergic neurons, *Slc17a7* (also known as *Vglut1*), *Slc17a6* (*Vglut2*) and *Slc17a8* (*Vglut3*) were differentially distributed in different brain regions[28] (Fig. 2e, left). *Slc17a7* dominated in the olfactory areas, isocortex, hippocampal formation, cortical subplate and cerebellar cortex, whereas *Slc17a6* dominated in the hypothalamus, midbrain and hindbrain. In some regions, *Slc17a7* and *Slc17a6* were co-expressed in neurons, such as the retrosplenial areas, pontine grey, anterior olfactory nucleus and thalamus (Fig. 2e, left, and Extended Data Fig. 4i). The less used *Slc17a6* was scattered across multiple regions, enriched in regions such as layer 5 of the isocortex and the bed nuclei of the stria terminalis, and was often co-expressed with *Slc17a7* and/or *Slc17a6* (Fig. 2e, left).

We also located the neurons that used other, modulatory neurotransmitters (Fig. 2c,e, right). Dopaminergic neurons were observed in the olfactory areas (located in the glomerular layer), hypothalamus (enriched in the arcuate hypothalamic nucleus) and midbrain (enriched in the ventral tegmental area and neighbouring areas)[29] (Extended Data Fig. 4j). Serotonergic neurons were enriched in the raphe nuclei (dorsal nucleus raphe, nucleus raphe pontis and nucleus raphe magnus) in the midbrain and hindbrain[30] (Extended Data Fig. 4k). Histaminergic neurons were observed in the ventral tuberomammillary nucleus, tuberal nucleus and other neighbouring areas in the ventral hypothalamus[31] (Extended Data Fig. 4l). Glycinergic neurons were widely distributed across the hindbrain[32] (Extended Data Fig. 4m). Noradrenergic neurons were localized to the locus coeruleus and neighbouring areas in the hindbrain[33] (Extended Data Fig. 4n). Cholinergic neurons were widely distributed in the striatum, ventral pallidum and multiple small subregions such as the medial habenula in the thalamus, the arcuate hypothalamic nucleus in the hypothalamus, the parabigeminal nucleus in the midbrain and the dorsal motor nucleus of the vagus nerve in the hindbrain[34] (Extended Data Fig. 4o).

These modulatory transmitter transporter genes were often co-expressed with glutamate or GABA transporters in individual neurons. For example, dopaminergic neurons in the olfactory areas co-expressed *Slc32a1*, and co-expression with *Slc32a1* or *Slc17a6* were both observed in the midbrain and hypothalamus. Cholinergic neurons in the striatum and pallidum co-expressed *Slc32a1* and those in the hindbrain also co-expressed *Slc17a6*. Glycinergic neurons and histaminergic neurons co-expressed *Slc32a1*.

Our MERFISH data also showed spatially heterogeneous distributions of many neuropeptide genes (Extended Data Fig. 5). To name just a few examples: *Adcyap1* and *Gal* were enriched in multiple nuclei in the hypothalamus; *Penk* was widely expressed in the striatum, midbrain and cerebellum, and particularly enriched in the striatum; and *Tac2* was enriched in the bed nuclei of the stria terminalis and multiple nuclei in the hypothalamus, striatum and thalamus.

We also observed two subclasses of immature neurons (IMNs): one inhibitory and one excitatory (Fig. 2f, left). The inhibitory IMNs, composed of 30 clusters, were distributed along the subventricular zone (SVZ), extending to the olfactory bulb through the anterior commissure (Fig. 2f, middle, and Extended Data Fig. 4p), consistent with previous findings of adult neurogenesis in the SVZ and migration of the neuroblast to the olfactory bulb along the rostral migratory stream[35–37]. The excitatory IMNs, composed of seven clusters, were found in two distinct locations: cluster 516 was primarily located in the piriform area of the olfactory areas, whereas the other clusters were distributed along the dentate gyrus in the hippocampal formation (Fig. 2f, right), consistent with previous findings of adult neurogenesis in the hippocampal formation[38,39].

## Diversity and spatial organization of non-neuronal cells

The non-neuronal cells comprised 23 subclasses and 117 clusters (Fig. 3a and Supplementary Table 2). We quantified the non-neuronal cell-type composition and enrichment in the 11 major brain regions, as well as in fibre tracts and ventricular systems where non-neuronal cells dominate (Fig. 3b,c and Supplementary Table 2). Across the whole brain, non-neuronal cells were composed of 30% of oligodendrocytes, 6% of oligodendrocyte progenitor cells (OPCs), 28% of vascular cells (endothelial cells, pericytes, vascular leptomeningeal cells (VLMCs), smooth muscle cells (SMCs) and arachnoid barrier cells), 23% of astrocytes, 8% of immune cells (microglia, border-associated macrophages (BAMs), lymphoid cells, dendritic cells and monocytes) and 5% other cell types (olfactory ensheathing cells, Bergmann cells, ependymal cells, choroid plexus cells, tanycytes and hypendymal cells) (Fig. 3b).

Of note, some non-neuronal cell types also exhibited strong regional specificity, especially for astrocytes and cells in the ventricular systems (Fig. 3c). We observed a high diversity of astrocytes, including 36 cell clusters. Among these, the two biggest clusters, Astro 5225 and Astro 5214, accounted for 48% and 33% of the total astrocyte population, respectively. Astro 5225 was exclusively located in the telencephalon and Astro 5214 in non-telencephalic regions (Fig. 3d), consistent with previous observation[16]. In addition, Astro 5215 and 5216 were located in the thalamus and hindbrain, respectively; Astro 5231–5236 were located in the olfactory bulb; Astro 5207 was located in the cerebellum; Astro 5222 was located in the dentate gyrus; Astro 5208 was enriched in the medulla close to the pia surface; and Astro 5228, 5229 and 5230 were located along the SVZ, extending to the olfactory bulb, and were colocalized extensively with the inhibitory immature neurons (Fig. 3d). The locations of Astro 5228–5230 were consistent with previous observations that the migratory steam of neuroblasts generated in the SVZ are ensheathed by cells of astrocytic nature[35–37,40]. Although not all enumerated here, essentially every Astro cluster showed unique spatial distributions (Fig. 3d). The Astro-like Bergmann cells were located in the cerebellum (Fig. 3d).

Oligodendrocytes were enriched in the fibre tracts and were highly abundant throughout the brain stem, whereas OPCs were evenly distributed across the whole brain (Fig. 3e). At the cluster level, some oligodendrocytes and OPCs also showed regional specificity. For example, Oligo 5277 was enriched in the cortex, whereas Oligo 5286 was enriched in the hindbrain (Fig. 3e).

We also observed region-specific distribution of the cells related to the ventricular systems. In the third ventricle, tanycytes resided in the ventral region, whereas ependymal cells occupied the dorsal region (Fig. 3f), consistent with previous work[41,42]. Hypendymal cells were located in the subcommissural organ at the dorsal third ventricle (Fig. 3f). The primary residents inside the ventricles were choroid plexus cells and VLMCs (Fig. 3f). Most VLMC clusters were restricted to

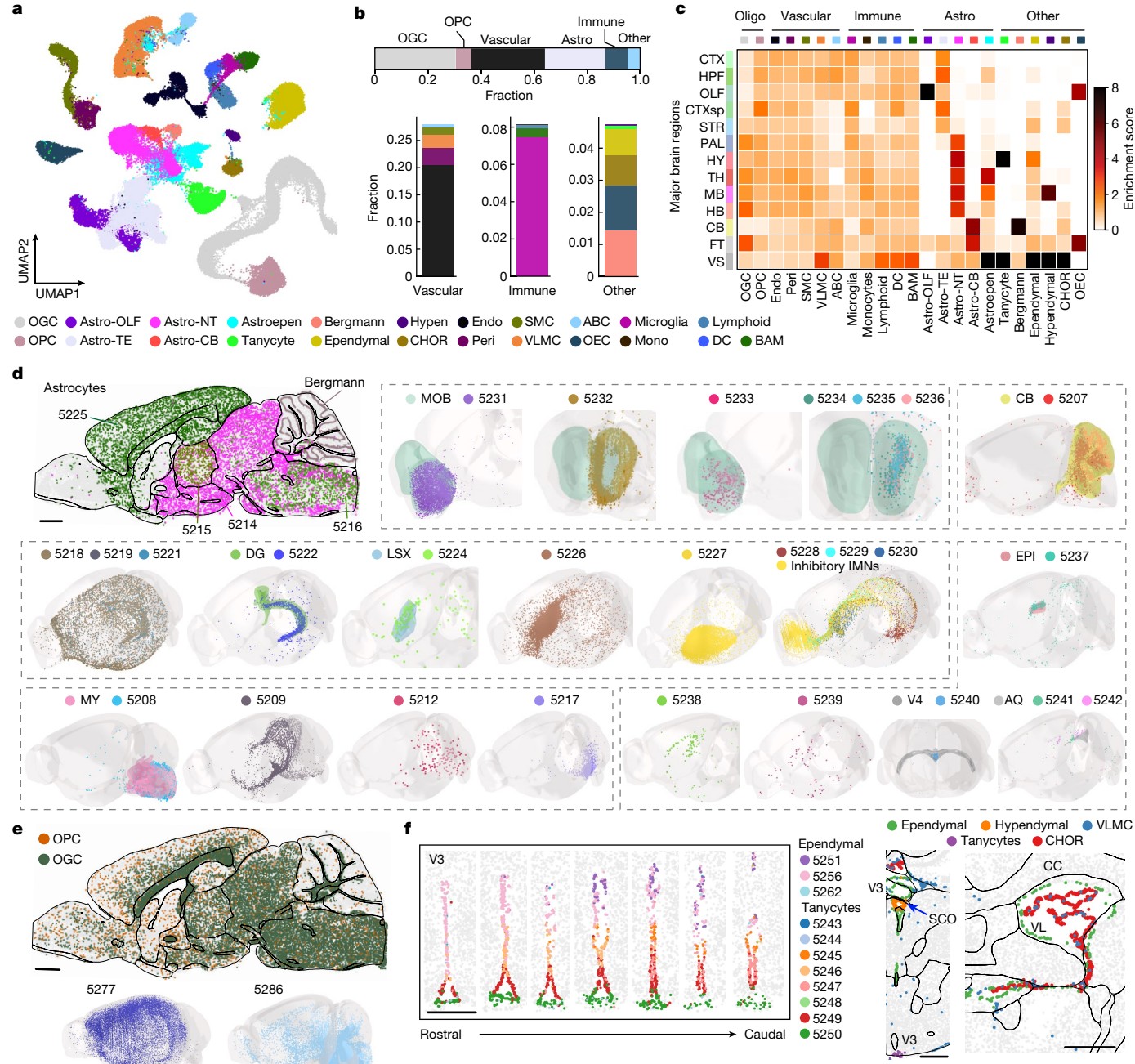

**Fig. 3 | Cell-type compositions and spatial distributions of non-neuronal cells. a**, UMAP of non-neuronal cells coloured by subclass identities as shown in the legend. ABC, arachnoid barrier cell; astro, astrocyte; CHOR, choroid plexus epithelial cell; DC, dendritic cell; endo, endothelial cell; hypen, hypendymal cell; mono, monocytes; NT, non-telencephalic; OEC, olfactory ensheathing cell; OGC, oligodendrocyte; peri, pericytes; TE, telencephalic. Astro-OLF, Astro-TE, Astro-NT and Astro-CB are the subclasses of astrocytes located in the olfactory areas, telencephalic regions, non-telencephalic regions and cerebellum, respectively. **b**, Bar plots showing the fractions of major non-neuronal cell types in the whole brain (top). Fractions of different vascular cell types, immune cell types and non-neuronal cell types in the 'other' category with cell subclasses coloured as shown in the legend. **c**, Heatmap showing the enrichment scores of all non-neuronal subclasses in 11 major brain regions, as well as in fibre tracts and ventricular systems. The enrichment score is defined as in Fig. 2b. **d**, Spatial distributions of the 31 astrocyte clusters, which contained more than 50 cells

(out of the 36 astrocyte clusters in total) and Bergmann cells, shown in a sagittal section (top left) and in the 3D CCF space (other panels), with cells coloured by cluster identities and cluster numerical indices. AQ, cerebral aqueduct; EPI, epithalamus; LSX, lateral septal complex; MY, medulla; V4, fourth ventricle. **e**, Spatial distributions of the OGCs and OPCs shown in a sagittal section with cells coloured by subclass identities (top). Two clusters are shown in the 3D CCF space (bottom). **f**, Spatial maps of three ependymal and eight tanycyte clusters in the third ventricle (V3) in seven coronal sections, 100 μm apart from each other along the rostral–caudal direction (left). Spatial maps of CHORs, ependymal cells, hypendymal cells and VLMCs in the third ventricle and lateral ventricle (VL) (right). Scale bars, 1 mm (**d**,**e**) and 0.5 mm (**f**). CC, corpus callosum; SCO, subcommissural organ. The underlying contour lines marking brain region boundaries in **d**–**f** and the 3D brain contours in **d** and **e** were generated using coordinates from the Allen Mouse Brain CCFv3 (ref. 21).

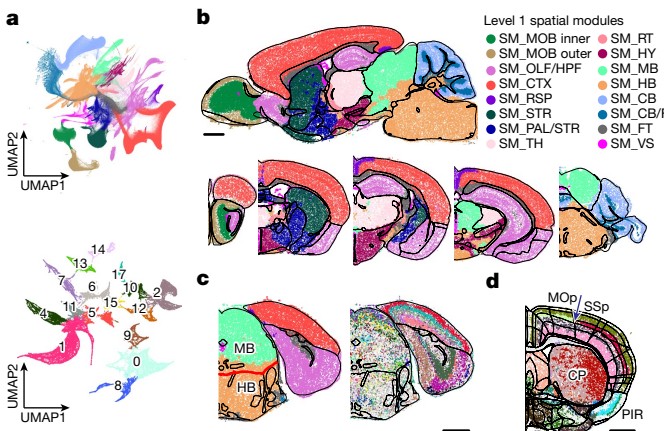

**Fig. 4 | Spatial modules: molecularly defined brain regions. a**, UMAP visualization of spatial modules. For any given cell, a local cell-type composition vector is calculated and used to cluster cells to determine the spatial modules (Methods). Level 1 spatial modules are determined with the cell-type composition determined at the subclass level; level 2 spatial modules are then determined for each level 1 spatial module with cell-type composition determined at both the subclass and the cluster levels and with only neurons considered. UMAP of cells in local cell-type composition space with cells coloured by their level 1 spatial module identities (top). UMAP of cells in one of the level 1 spatial modules (SM_TH, located at the thalamus) with cells coloured by their level 2 spatial module identities (bottom). **b**, Spatial maps of cells, coloured by their level 1 spatial module identities, shown in example sagittal and coronal sections. RSP, retrosplenial area; RT, reticular nucleus of the thalamus. **c**, Spatial maps of cells in one coronal section coloured by level 1 spatial module identities (left) and by cell subclass identities (right). The black lines mark the major brain region boundaries defined in the Allen CCF, and the CCF boundary between the midbrain and hindbrain is highlighted in red. **d**, Spatial map of cells coloured by level 2 spatial module identities in one coronal section. The black lines mark major brain region boundaries, and the thin grey lines mark the subregion boundaries defined in the CCF. The boundary between the primary motor cortex (MOp) and the primary somatosensory cortex (SSp) is indicated by the blue arrow. Scale bars, 1 mm (**b**–**d**). CP, caudoputamen. The underlying contour lines marking brain region boundaries in **b**–**d** were generated using coordinates from the Allen Mouse Brain CCFv3 (ref. 21).

pia, except for two distinct types: VLMC 5301 was enriched in the grey matter, and VLMC 5302 was located in the choroid plexus in the lateral and fourth ventricles (Fig. 3f and Extended Data Fig. 6a). ABCs resided in the subarachnoid space (Extended Data Fig. 6b). Other vascular cells (endothelial cells, pericytes and SMCs), which outline blood vessels, were broadly distributed (Extended Data Fig. 6c). Likewise, immune cells were also scattered across the brain (Extended Data Fig. 6d). As expected, olfactory ensheathing cells were located at the periphery of the olfactory bulb (Extended Data Fig. 6e).

## Molecularly defined brain regions (spatial modules)

The comprehensive spatial distributions of the transcriptionally distinct cell populations allowed us to construct a map of molecularly defined brain regions. To this end, we defined for each cell a local cell-type composition vector and clustered the cells using these vectors (Methods), resulting in 'spatial modules' that contained cells with similar neighbourhood cell-type compositions. We identified 16 level 1 spatial modules and 130 level 2 spatial modules (Fig. 4a, Extended Data Fig. 7 and Supplementary Table 3).

Level 1 spatial modules segmented the brain into areas that largely coincided with the major brain regions defined in the CCF (Fig. 4b). One notable discrepancy was the boundary between the midbrain and the hindbrain (Fig. 4c). This discrepancy originated from the gradual changes of cell-type compositions from the midbrain to the hindbrain,

making an unambiguous determination of the midbrain–hindbrain boundary challenging. At level 2, many spatial modules were consistent with the subregions defined in the CCF, but we observed more discrepancies (Fig. 4d) due to multiple possible reasons. On the one hand, our spatial module delineation was based on cell types defined by transcriptome-wide expression profiles of individual cells and hence have a higher molecule resolution than the information used in brain region delineation in the CCF. For example, our analysis segmented the caudoputamen into a lateral and medial spatial module, whereas such division is not shown in the CCF (Fig. 4d). In fact, a spatial gradient represents a more precise description of the molecular profile of this region, as described in the next section. On the other hand, we also noticed that some subregion boundaries defined by connectional or functional information in the CCF were missing in the transcriptionally defined spatial modules. For example, the isocortex is divided into multiple subregions in the CCF, whereas such boundaries were largely missing in the spatial module analyses except for the boundary between the primary motor cortex and the primary somatosensory cortex in layer 4 (Fig. 4d).

## Spatial gradients of molecularly defined cell types

Although clustering algorithms group cells into discrete spatial modules or cell types, the gene expression profiles of cells may exhibit a gradual or continuous change in some cases. Indeed, the coexistence of discrete and continuous cell-type heterogeneity has been previously observed in multiple brain regions[8,43–47], with some continuous cellular heterogeneity forming a gradient along a spatial direction[8,45–47].

We thus examined all cell subclasses to identify the spatial gradients of cells, in which the gene expression of cells changed gradually in space. To this end, we quantified the discreteness of clusters within each subclass and observed that most of the subclasses contained continuously connected cell clusters (Methods and Extended Data Fig. 8a). We further identified the cell subclasses that exhibited a prominent spatial axis along which the gene expression profiles of cells changed gradually, representing a spatial gradient, using the pseudotime[8,48] or the first principal component (PC1) to quantify gene expression changes. Moreover, to capture the gradients that spanned multiple subclasses, we assessed whether the gradients identified within subclasses extended into transcriptionally similar subclasses.

We identified many spatial gradients in different brain regions. For example, IT neurons formed a continuous gradient across the whole isocortex, where the gene expression changed gradually along the cortical depth direction but with a more discernible separation for the layer 2/3 IT neurons (Fig. 5a), consistent with our previous results for the primary motor cortex[8]. In the striatum, D1 and D2 medium spiny neurons both formed a spatial gradient along the dorsolateral–ventromedial axis (Fig. 5b,c), consistent with previous observations[45]. In the lateral septal complex, several GABAergic subclasses formed a gradient along the dorsoventral axis (Fig. 5d). Spatial gradients were also observed in the CA1, CA3 and dente gyrus regions of the hippocampus (Extended Data Fig. 8b–d) and in the inferior colliculus of the midbrain (Extended Data Fig. 8e). We also observed spatial gradients among some non-neuronal cells. For example, tanycytes formed a continuous gradient along the dorsoventral axis of the third ventricle (Fig. 5e). Overall, spatial gradients of cells were widespread in many brain regions.

We also noticed a large-scale spatial gradient spanning the hypothalamus, midbrain and hindbrain regions. Here we visualized the gradient in the gene expression uniform manifold approximation and projection (UMAP), where each neuron was coloured by its spatial coordinates (Fig. 5f). An overall rostral–caudal gradient of gene expression change from the hypothalamus to the midbrain and then the hindbrain, as well as a dorsal–ventral gradient from the midbrain to the hypothalamus and hindbrain, were observed in the UMAP.

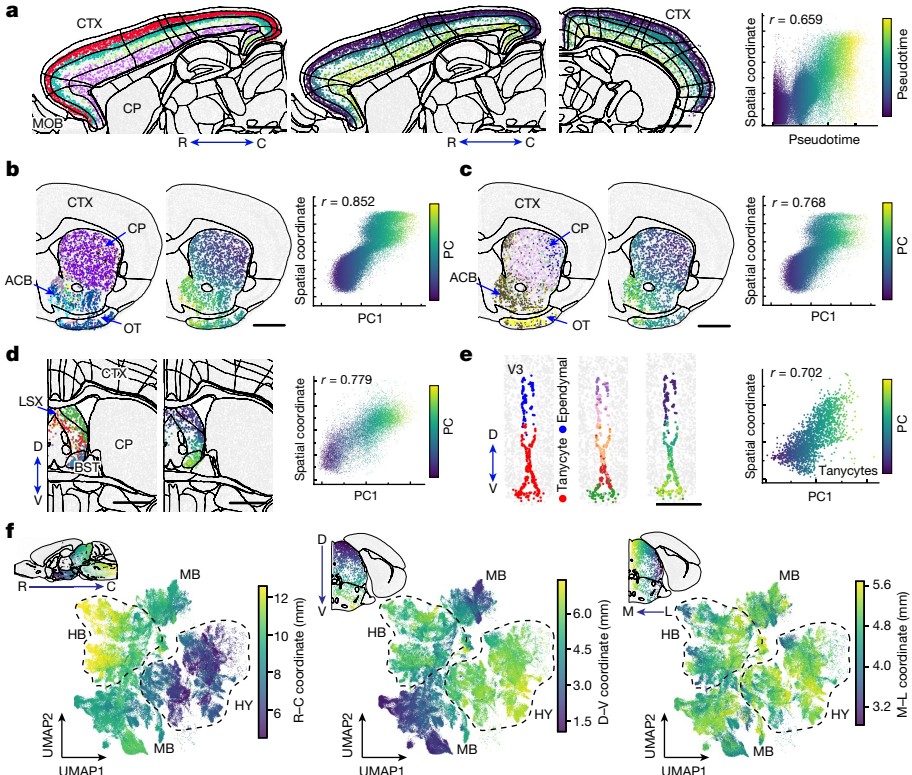

**Fig. 5 | Spatial gradients of molecularly defined cell types. a**, Spatial gradient of IT neurons in the isocortex. From left to right: spatial map of IT neurons coloured by subclass identities in a sagittal section; spatial maps of IT neurons coloured by pseudotime in the same sagittal section and an additional coronal section; and a correlation plot of pseudotime versus cortical depth for individual IT neurons, coloured by pseudotime values. The Pearson correlation coefficient *r* is shown. C, caudal; R, rostral. **b**, Spatial gradient of the D1 medium spiny neurons (STR D1) in the striatum. From left to right: a spatial map of STR D1 neurons coloured by subclass identities in a coronal section; a spatial map of STR D1 neurons coloured by the first principal component (PC1) in the same coronal section; and a correlation plot of PC1 value versus spatial coordinate for individual STR D1 neurons, coloured by PC1 values. ACB, nucleus accumbens;

OT, olfactory tubercle. **c**–**e**, Same as **b** but for spatial gradients of STR D2 neurons in the striatum (**c**), GABAergic neurons in the LSX (**d**) and tanycytes in the third ventricle (V3) (**e**). BST, bed nuclei of the stria terminalis; D, dorsal; V, ventral. **f**, Large-scale gradient of neurons across the hypothalamus, midbrain and hindbrain. The UMAPs were generated based on the gene expression profiles of neurons, and individual cells are coloured by their spatial coordinates along the rostral–caudal (left), dorsal–ventral (middle) and medial (M)–lateral (L) (right) axes. The insets show example brain slices with cells in the regions of interest coloured by the relevant spatial coordinates. Scale bars, 1 mm (**a**–**d**) and 0.5 mm (**e**). The underlying contour lines marking brain region boundaries in **a**–**d** and **f** were generated using coordinates from the Allen Mouse Brain CCFv3 (ref. 21).

## Cell-type-specific cell–cell interactions and communications

The high-resolution spatial atlas of molecularly defined cell types further allowed us to infer cell-type-specific cell–cell interactions or communications arising from soma contact, paracrine signalling or other short-range interactions. Here we considered cell types at the subclass level and inferred cell-type-specific cell–cell interactions in individual brain regions by querying whether the soma contact or proximity frequency observed between a given cell-type pair was higher than random chance, supplemented with expression variation analysis of ligand–receptor pairs (Methods and Fig. 6a). We determined the random chance (null distribution of probability) by performing local spatial-coordinate randomizations to disrupt the spatial relationship between neighbouring cells while preserving the local density of each cell type and hence brain structures[11]. We identified several hundred pairs of cell subclasses showing statistically significant interactions by our criteria (Fig. 6b,c, Extended Data Fig. 9 and Supplementary Table 4). Most of our predicted interacting cell-type pairs contained multiple ligand–receptor pairs that showed significant upregulation in expression in the proximal cell pairs compared with non-proximal cell pairs within the same cell-type pair (Supplementary Table 5), providing insights into the molecular basis of these cell–cell interactions.

Our predicted cell–cell interactions included interactions among non-neuronal cells, between non-neuronal cells and neurons, and among neurons. Below, we describe examples in each of these three categories. As examples in the first category, we observed interactions between vascular cells and immune cells. Both endothelial cells and pericytes showed significant interactions with BAMs, macrophages in the brain (Fig. 6d,e). In both cases, ligand–receptor pairs from the laminin signalling pathway showed significant upregulation in the proximal cell pairs compared with non-proximal cell pairs (Fig. 6d,e). Laminins at the endothelial basement membrane can promote monocyte differentiation to macrophages[49]. Thus, these cell–cell interactions might have a role in regulating the pool of macrophages in the brain. We also observed significant interactions between microglia and these two vascular cell types (Fig. 6f). Compared with endothelial cells, pericytes exhibited a higher probability to interact with microglia, whereas an opposite trend was observed for their interactions with BAMs (Fig. 6g).

We also observed significant interactions between neurons and non-neuronal cells. For example, astrocytes and inhibitory IMNs showed significant interactions in the olfactory bulb (Extended Data Fig. 10a). Neuroblasts migrating from the SVZ to the olfactory bulb interact with cells of astrocytic nature along the rostral migratory stream[35–37,40]. Whether our observed IMN–astrocyte interactions in the olfactory bulb is related to the interactions between neuroblasts and astrocytes in the rostral migratory stream remains an open question.

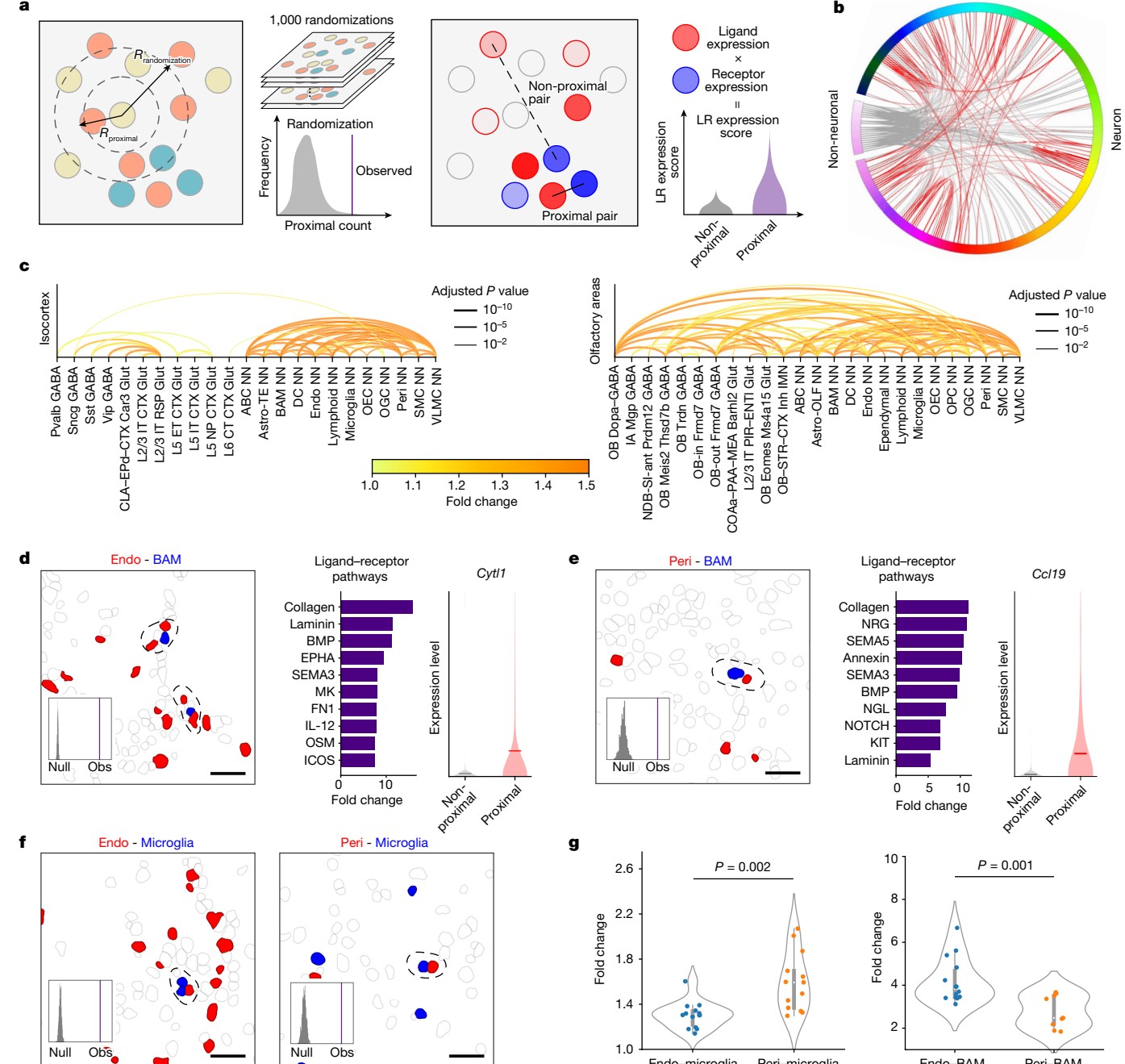

**Fig. 6 | Cell–cell interactions and communications. a**, Schematics of cell–cell interaction analysis (left) and ligand–receptor (LR) analysis (right). $R_{proximal}$ denotes the proximity distance threshold; two cells are considered in contact or proximity if the distance between their centroid positions is within this distance threshold. $R_{randomization}$ denotes the randomization radius; we shifted the spatial location of each cell to a random position within $R$ from its original location to generate the null distribution. **b**, Cell–cell interactions across the whole brain. Each line corresponds to a predicted interacting cell-type pair. The grey lines indicate interactions between non-neuronal cells and neurons or among non-neuronal cells; the red lines indicate neuron–neuron interactions. **c**, Cell–cell interactions in two brain regions. Each line corresponds to an interacting cell-type pair, with the colour indicating fold change in proximity frequency compared with random chance and thickness indicating *P* values corrected by the Benjamini–Hochberg procedure. CLA, claustrum; CT, cortical-thalamic; EPd, endopiriform nucleus, dorsal part; ET, extratelencephalic; IT, intratelencephalic; L2/3, layer 2/3; NN, non-neuronal; NP, near-projecting. **d**, Interactions between endothelial cells and BAMs. Example image of cells, with cells of the indicated cell types shown in red and blue and all other cells shown in grey (left). Proximal cell pairs are circled by a

dashed line. Observed counts (Obs) of the proximal cell pairs and the null distributions (null) from randomization control are shown in the inset. Top 10 ligand–receptor pathways upregulated in proximal cell pairs as compared to non-proximal cell pairs (middle). When multiple ligand–receptor pairs in a pathway are upregulated, the plotted fold-change value represents that of the pair with the highest upregulation fold change. Expression distributions of the indicated gene in endothelial cells proximal (red) or non-proximal (grey) to BAMs (right). Scale bar, 30 μm. Horizontal lines in the violin plots indicate median. **e**, Same as **d**, but for interactions between pericytes and BAMs. **f**, Interactions between endothelial cells and microglia (left) and between pericytes and microglia (right). **g**, Fold changes of observed proximal cell-pair number relative to the null-distribution mean across different brain regions. Each data point represents a brain region where significant interactions were observed (*P* values were calculated by two-sided Welch's *t*-test; the centre points indicate the median, the boxes denote the interquartile range and the whiskers indicate 1.5 times the interquartile range). Comparison between endothelial–microglia and pericyte–microglia interactions (left), and comparison between endothelial–BAM and pericyte–BAM interactions (right).

We also observed significant interactions between astrocytes and excitatory IMNs in the hippocampal formation (Extended Data Fig. 10b). Many additional astrocyte–neuron interactions were observed across various brain regions (Fig. 6c and Extended Data Fig. 9). Many astrocyte–neuron interactions may also be missed in our analysis because astrocytes often interact with neurons through their processes instead of cell bodies.

Although not designed to capture long-range synaptic communications between neurons, our analyses also predicted interactions between some neuronal subclasses, for example, between Pvalb chandelier GABA neurons and CA3 glutamatergic neurons in the hippocampal formation (Extended Data Fig. 10c) and between IPN Otp Crisp1 GABA neurons and DTN–LDT–IPN Otp Pax3 GABA neurons in the midbrain (Extended Data Fig. 10d). The proximal pairs of chandelier neurons and CA3 glutamatergic neurons showed pronounced upregulation of ligand–receptor pairs in the WNT pathways (Extended Data Fig. 10c). WNT signalling is known to be important for hippocampal functions[50], as well as dysfunction in neurological disorders, such as spatial memory impairment and anxiety-like behaviour[51]. Chandelier neurons and CA3 glutamatergic neurons have also been implicated in these neurological disorders[52,53]. Whether our observed interactions between chandelier and CA3 glutamatergic neurons are involved in these disorders awaits future investigations.

Given the importance of WNT signalling in brain development, function and diseases, we performed a systematic quantification of various WNT ligands in cell–cell interactions in different brain regions. Interacting non-neuronal cells primarily showed upregulation of a subset of WNT ligands, *Wnt4*, *Wnt5a*, *Wnt5b*, *Wnt6* and *Wnt9a*, across nearly all brain regions (Extended Data Fig. 10e, top). Conversely, the usage of WNT signalling in neuron–neuron and neuron–non-neuronal cell communications showed high regional specificity, as well as WNT ligand specificity (Extended Data Fig. 10e, middle and bottom). Overall, among the ligand–receptor pairs that we observed to be upregulated in interacting cells in the brain, WNT, laminin, collagen, semaphoring and BMP-related pathways were among the most broadly used (Extended Data Fig. 10f).

In addition to ligands and receptors, we also identified other genes that were upregulated in the predicted interacting cell pairs (Supplementary Table 6), which suggest potential functional roles of these cell–cell interactions. We illustrate this with examples in the non-neuronal–non-neuronal, neuronal–non-neuronal and neuronal–neuronal interaction categories. For example, some cytokines were upregulated in vascular cells proximal to BAMs (for example, *Cytl1* in endothelial cells and *Ccl19* in pericytes) (Fig. 6d,e). These cytokines have been shown to be chemoattractants for macrophages[54,55]. Our observations suggest the possibility that vascular cells in the brain may use these cytokines to recruit macrophages. As another example in the first category, genes involved in elastic fibre assembly, including *Eln*, *Fbln2* and *Fbln5*, were significantly upregulated in endothelial cells proximal to SMCs (Extended Data Fig. 10g), consistent with previous findings that endothelial cells make elastic fibres that inhibit the growth of SMCs[56]. We further observed that *Pi16* was significantly upregulated in endothelial cells proximal to SMCs (Extended Data Fig. 10g). *Pi16* can inhibit the growth of cardiomyocytes[57]. We thus hypothesize that *Pi16* expressed by endothelial cells may be a growth inhibitor of SMCs in the brain. As an example in the second category – interactions between neurons and non-neuronal cells – we observed that *Sfrp1*, a WNT signalling modulator, was upregulated in astrocytes proximal to inhibitory IMNs in the olfactory bulb (Extended Data Fig. 10a). *Sfrp1* expressed in OPCs can inhibit the proliferation of neural stem cells[58]. Our results suggest the possibility that astrocytes may use *Sfrp1* to modulate WNT signalling and regulate adult neurogenesis. Finally, as an example in the neuronal–neuronal interaction category, we observed that the glutamate receptor GRIN2A was upregulated in parvalbumin-positive chandelier neurons proximal to CA3 glutamatergic neurons (Extended Data Fig. 10c), suggesting the possibility that communications between these neurons may affect the synaptic properties of chandelier neurons.

## Discussion

In this work, we generated a comprehensive atlas of molecularly defined cell types across the whole mouse brain with high molecular and spatial resolution. By imaging approximately 10 million cells with MERFISH and integrating the MERFISH data with a whole-brain scRNA-seq dataset, we determined the spatial organization of more than 5,000 transcriptionally distinct cell clusters, which were grouped into 338 cell subclasses, and imputed a transcriptome-wide expression profile for each imaged cell. We further registered this atlas to the Allen Mouse Brain CCF, providing a reference cell atlas that can be broadly used by the scientific community. This CCF registration allowed us to determine the composition, spatial organization and potential interactions of transcriptionally distinct cell types in each individual brain region.

Our results highlight an extraordinary molecular diversity and spatial heterogeneity of neurons. We observed more than 5,000 transcriptionally distinct neuronal cell clusters belonging to 315 subclasses. At the subclass level, individual cell types exhibited strong enrichment, if not located exclusively, within one of the 11 major brain regions. At a finer scale, most transcriptionally distinct neuronal clusters within individual subclasses also adopted different spatial distributions from each other. Telencephalic regions (the olfactory areas, isocortex, hippocampal formation, cortical subplate, striatum and pallidum) showed lower cellular diversity than that observed in the hypothalamus, midbrain and hindbrain, which contained a substantially larger number of transcriptionally distinct cell populations in each region. Moreover, cells in these latter regions exhibited complex spatial organization with transcriptionally distinct cell types often assuming irregularly shaped, partially overlapping spatial distributions, whereas spatial organization of cells showed a higher level of regularity in the telencephalic regions, such as the layer-specific distribution of cortical neurons. The comprehensive mapping of spatial distributions of the transcriptionally distinct neuronal cell types allowed us to partition the brain into molecularly defined brain regions, which we termed spatial modules. We also observed many spatial gradients in the brain where the cell-type composition and molecular profiles of cells change gradually in space.

Our data also provide a systematic molecular and spatial characterization of the non-neuronal cells. Non-neuronal cells accounted for about half of the cells in the adult mouse brain, and this fraction varied substantially from region to region. We observed a high diversity of non-neuronal cells, comprising 117 transcriptionally distinct clusters belonging to 23 subclasses. Of note, many non-neuronal cell types also exhibited a highly level of regional specificity. This spatial heterogeneity was particularly pronounced for astrocytes, with each astrocyte cluster adopting a unique spatial distribution. Although such regional-specific molecular profiles of astrocytes likely have a developmental origin, it is possible that the interactions of astrocytes with distinct types of neurons in different brain regions also contribute to the molecular diversity of astrocytes. An interesting question arises as to whether the different molecular properties of astrocytic subtypes have an important role in their function to support and modulate the activity of diverse neuronal cell types.

Our high-resolution cell atlas further enabled a brain-wide investigation of cell-type-specific cell–cell interactions or communications. We predicted interactions or communications between several hundred pairs of cell types at the subclass level. Our analysis of ligand–receptor pairs, as well as other genes, upregulated in proximal cell pairs within each of these cell-type pairs further suggest potential molecular basis and functional roles of these cell–cell interactions. Although the combination of spatial and molecular information in MERFISH data offers unique advantages in predicting cell–cell interactions or communications, a few factors could still cause false positives and

negatives in our analyses. On the false-positive side, although we used local spatial randomizations of cells to generate null distributions to reduce the confounding effect of colocalization of cell types in a brain structure without interactions, and we further imposed the requirement of ligand–receptor upregulation in proximal cell pairs in cell–cell interaction calling, it is impossible to completely eliminate such a confounding effect, especially when colocalization occurs within a small brain structure. In addition, our requirement of ligand–receptor upregulation in a proximal cell pair, as compared with non-proximal cell pairs, for cell–cell interaction calling could also cause false negatives, because ligand–receptor pairs mediating interactions between two cell types may be expressed at a constant level regardless of cell–cell proximity. One could adjust the parameters and requirements in our analysis to generate a more stringent or a more inclusive list of cell–cell interaction hypotheses. Regardless of the parameter choice, additional experiments are needed to validate these cell–cell interaction hypotheses.

Overall, our data provide a molecularly defined and spatially resolved cell atlas of the entire adult mouse brain, featuring complex organizations of thousands of distinct cell populations. This reference cell atlas provides a foundation for future functional studies of these distinct cell populations. Both the molecular signatures and the spatial information in the atlas provide handles for functional interrogation of specific neuronal cell types through transgenic targeting tools and optogenetic manipulations. In addition, the predicted interactions between non-neuronal cells and neuronal cells and among non-neuronal cells, as well as the associated upregulation of ligand–receptor pairs and other genes, provide hypotheses and entry points for testing the functional roles of the diverse non-neuronal cell types. Furthermore, the combination of transcriptomic imaging with neuronal activity imaging under various behaviour paradigms[4,5,59,60] can reveal the functional roles of neurons. We envision that future studies combining spatially resolved transcriptomic analysis with measurements of various other properties, such as epigenomic profiles, morphology, connectivity and function of cells, as well as with systematic gene perturbation methods, will help to connect our understanding of the molecular and cellular architecture of the brain with its function and dysfunction in health and diseases.

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

## Methods

### Animals

Adult C57BL/6NCrl (strain code: 027, Charles River Laboratories) male and female mice 56–62 days of age were used in this study. Animals were purchased at an age that was 1 week younger (49–55 days) than the target age for euthanasia and were housed at Harvard University Animal Facility for 1 week to acclimate before being killed. Mice were maintained on a 12 h–12 h light–dark cycle (14:00 to 2:00 dark period) with at a temperature of 22 ± 1 °C, a humidity of 30–70%, with ad libitum access to food and water. All animals used in this study were killed between 14:00 and 18:00 of the day. Animal care and experiments were carried out in accordance with US National Institutes of Health guidelines and were approved by the Harvard University Institutional Animal Care and Use Committee.

### Bulk RNA-seq of the whole mouse brain

Estimates of the average RNA expression levels of individual genes in the mouse brain were derived from the bulk RNA-seq data of the whole mouse brain. RNA was extracted and isolated using the RNAqueous Micro total RNA isolation kit (AM1931, Thermo Fisher) following the manufacturer's instructions from three different whole mouse brains 56–62 days of age. RNA quality was assessed using Agilent TapeStation and samples with an RNA integrity score of more than 8 were kept for sequencing. RNA-seq libraries were constructed using the Kapa mRNA HyperPrep kits and were sequenced using the Illumina NextSeq500 platform performed by the Bauer Center Sequencing Core at Harvard University.

### Single-cell RNA-seq data of the whole mouse brain

Single-cell RNA-seq data were generated by the Allen Institute (see companion manuscript by Yao et al.[18] in this BICCN package). These data are available at the Neuroscience Multi-omics Archive under identifier: https://assets.nemoarchive.org/dat-qg7n1b0.

### Gene selection for MERFISH

To discriminate transcriptionally distinct cell populations with MER-FISH, we designed the gene panels based on differentially expressed gene analysis using the scRNA-seq data. Genes differentially expressed between pairs of transcriptionally distinct cell clusters from the scRNA-seq data were selected based on the following criteria: the genes had twofold change or more in expression between the two clusters with $P < 0.01$; they were expressed in at least 50% cells in the foreground cluster, with more than 3.3-fold enrichment, in terms of the fraction of cells expressing the gene, relative to the background cluster. The top 50 genes that satisfied the criteria and ranked by $P$ values in each direction for every cell cluster pair were pooled together as the differentially expressed gene candidates for the final marker gene set. We then trimmed this differentially expressed gene pool to remove the genes that were too abundant or too short and thus were potentially challenging for MERFISH imaging experiments. Specifically, we excluded the genes that can accommodate fewer than 40 hybridization probes (MERFISH-encoding probes) and thus were approximately less than 500 nt in length (neighbouring target regions for encoding-probe binding are allowed to overlap, as described below), or were expressed at an average of 3,000 counts in its highest expressing cell cluster as determined by the scRNA-seq data.

We further trimmed down the list of differentially expressed genes determined above based on the significance of these genes in neuroscience studies and their effectiveness in distinguishing different cell clusters. This selection process began with 123 subclass markers defined based on scRNA-seq clustering results. We then continued to add differentially expressed genes that fell into the categories of transcription factors, neuropeptides, G protein-coupled receptors, interleukins and secreted proteins, including 229 genes in total. Following this, we used a greedy search algorithm to iteratively add genes that had the most potent discriminative power in distinguishing pairs of cell clusters that were not adequately separated by the already chosen genes. This greedy search was concluded once there were at least three differentially expressed genes included for each pair of clusters in each direction, which in total added up to approximately 1,100 genes. Finally, we added some manually picked genes of interest, such as a few circadian clock genes, previously known non-neuronal cell-type marker genes, neurotransmitter-related genes and neuropeptide genes, among others, to form the final gene panels.

Two gene panels were used in the MERFISH experiments. The first panel contained 1,124 genes and was used for imaging most of the slices in animal 2, which was the animal that we imaged first. The second gene panel contained 1,147 genes and was used for imaging the remainder of slices of animal 2 and for imaging all other animals (animals 1, 3 and 4). These two gene panels are very similar to each other. Compared with the first gene panel, we added 25 manually picked genes in the second panel, including additional cell-type markers for non-neuronal cells, additional neurotransmitter-related genes and neuropeptide genes, and we also removed two genes (*Nrgn* and *Mag*) from the first gene panel. The two gene panels have 98% of the genes (1,122 genes) in common, and only the 1,122 common genes from both panels were used to integrate the MERFISH data with scRNA-seq data for cell-type classification. Historically, animal 2 was imaged first, and we made the changes in the gene panel after imaging the majority of tissue slices from this animal. As the 1,122 common genes were sufficient for cell-type classification (it allowed us to integrate MERFISH and scRNA-seq data and transfer cell-type labels of all 338 subclasses and approximately 99% of the 5,322 cell clusters from the scRNA-seq data to MERFISH data with high confidence), we decided to keep all data from the first imaged animal and used the 1,122 common genes that were present in the data from all animals for the cell-type classification purpose. The 25 genes that we have added to the second gene panel were mostly good marker genes for specific cell types and have been previously studied. Therefore, although not being used for cell-type classification, these 25 genes can provide useful information for people interested in these genes or the specific cell types that these genes mark.

In addition to the MERFISH gene panel, we also imaged four other genes (*Sst*, *Vip*, *Avp* and *Pmch*) that can accommodate fewer than 40 hybridization probes or were expressed at an average of more than 3,000 counts in its highest expressing cell cluster. These genes were imaged in two sequential rounds of two-colour FISH imaging, following the MERFISH run that imaged the 1,124-gene or 1,147-gene panel. These genes were included because they were classified as subclass markers based on the scRNA-seq data. These sequential genes were also not used in the integration of the MERFISH data and the scRNA-seq data for cell-type classification. In the experiments with the 1,124-gene panel, we further included *Fos* in one extra sequential FISH imaging round, whereas *Fos* was included in the 1,147-gene panel.

### Design and construction of MERFISH-encoding probes

Encoding probes for the MERFISH gene panels were designed as previously described[4]. We first assigned to each of the 1,124 genes in the first gene panel a unique binary barcode drawn from a 32-bit, Hamming-Distance-4, Hamming-Weight-4 codebook. This codebook also included 116 extra barcodes as 'blank' barcodes, which were not assigned to any genes, to provide a measure of the false-positive rate in MERFISH measurement. For the second 1,147-gene panel, the additional 25 genes were each randomly assigned a barcode from the 116 blank barcodes.

Each MERFISH-encoding probe contained one 30-nt target sequence that could specifically bind to a target gene and two 20-nt readout sequences. We designed a total of 32 readout sequences, each corresponding to 1 bit of the 32-bit MERFISH code. The collection of encoding probes designed to bind to each gene contained the four

readout sequences corresponding to the 4 bits that read '1' in the barcode of that gene. Each encoding probe contained two of the four 20-nt readout sequences that encode the specific barcode assigned to the gene. To design the target sequences in the encoding probes, we identified all possible 30-nt targeting regions within each target gene as previously described[61]. In brief, for each gene, we selected 30-nt target regions that had a GC fraction between 40% and 60%, a melting temperature within the range of 66–76 °C, and no homology longer than 15 nt to rRNAs or tRNAs. From the set of all possible 30-nt target regions for each gene, we selected 64 target regions randomly to construct encoding probes. For the transcripts that were not long enough to accommodate 64 non-overlapping target regions, we allowed these 30-nt targeting regions to overlap by as much as 20 nucleotides to increase the number of probes. We also allowed the minimum number of probes to be included to reduce to 40, the target regions to have a GC fraction between 30% and 70%, and a melting temperature within the range of 61–81 °C. Among the 1,147 genes, 7 genes had between 40 and 64 probes and the remaining genes had 64 probes.

In addition, we concatenated two PCR primers to each encoding probe sequence, the first comprising the T7 promoter, and the second being a random 20-mer designed to have no region of homology greater than 15 nt with any of the encoding probe sequences designed above, as previously described[61].

With the template encoding probe sequences designed above, we constructed the MERFISH probe set as previously described[4]. The template molecules were synthesized as a complex oligo pool (Twist Biosciences) and amplified as previously described[61].

Encoding probes for the four genes imaged using two rounds of sequential two-colour FISH were produced in the same manner, except that 48 targeting sequences were selected for each gene if possible, and one single unique readout sequence was concatenated with targeting sequences for each gene. The four readout sequences used here, one for each gene, were different from the 32 readout sequences used for the genes imaged in the MERFISH run. These probes were purchased from Integrated DNA Technologies (IDT).

The amplified encoding probes for the MERFISH run and encoding probes for the sequential two-colour FISH rounds were mixed for tissue staining.

## Design and construction of MERFISH readout probes

We used two readout probe schemes for the 32-bit MERFISH imaging plus the two sequential rounds of FISH imaging:

(1) Direct readout strategy with dye-conjugated readout probes complementary to the readout sequences, as described previously[8]: 36 readout probes were designed, each complementary to one of the 36 readout sequences. Each readout probes were conjugated to one of the two dye molecules (Alexa750 or Cy5) via a disulfide linkage. These readout probes were synthesized and purified by Bio-synthesis, stored in Tris-EDTA buffer, pH 8 (Thermo Fisher) at a concentration of 1 μM at −20 °C.

(2) Two-step readout strategy with oligonucleotide adaptors, as described previously[62]: first, 36 adaptor probes were designed, each consisting of a sequence complementary to one of the 36 readout sequences, concatenated by two additional common readout sequences, each for one colour channel. These adaptor probes were purchased from IDT, resuspended in Tris-EDTA buffer, pH 8 (Thermo Fisher) to a concentration of 1 mM and stored at −20 °C. Second, two dye-conjugated readout probes were designed, each complementary to one common readout sequence for a colour channel, and each were conjugated to one of the two dye molecules (Alexa750, Cy5 or Alexa647) via a disulfide linkage. These readout probes were synthesized and purified by IDT, stored in Tris-EDTA buffer, pH 8 (Thermo Fisher) at a concentration of 100 μM at −20 °C.

## Tissue preparation for MERFISH

Mice 56–62 days of age were euthanized with $CO_2$, and their brains were quickly harvested and frozen immediately in optimal cutting temperature compound (Tissue-Tek O.C.T.; 25608-930, VWR), and stored at −80 °C until sectioning. Frozen brains were sectioned at −18 °C on a cryostat (Leica CM3050S). A continuous set of 10-μm-thick slices were collected for imaging. For animal 1, 10-μm-thick serial coronal sections were collected from the anterior edge to the posterior edge of the brain and every tenth section was kept; for animal 2, the brains were sectioned similarly as for animal 1, but every twentieth coronal section was kept; for animal 3, 10-μm-thick serial sagittal sections were collected from the midline to the lateral edge of the brain and every twentieth section was kept; and for animal 4, the brains were sectioned similarly as for animal 3, but only the sections corresponding to the approximately same medial–lateral positions as the ones that showed broken regions for animal 3 were imaged. Each coverslip contained 2–4 coronal slices or 1–2 sagittal slices. In total, 150 slices were successfully imaged for animal 1, 67 slices were successfully imaged for animal 2, 25 slices were successfully imaged for animal 3, and 3 slices were imaged for animal 4. The coverslips were prepared as previously described[4].

Tissue slices were fixed by treating with 4% paraformaldehyde in 1× PBS for 15 min and were washed three times with 1× PBS and stored in 70% ethanol at 4 °C for at least 18 h to permeabilize cell membranes. The tissue slices from the same animal were sectioned at the same time and were stored in 70% ethanol at 4 °C for no longer than 2 months until all the tissue sections from the same animal were imaged.

The tissue slices were then stained with the MERFISH-encoding probes. In brief, the samples were removed from the 70% ethanol and washed with 2× saline sodium citrate (2× SSC) for three times. Then, we equilibrated the samples with encoding-probe wash buffer (30% formamide in 2× SSC) for 5 min at room temperature. The wash buffer was then aspirated from the coverslip, and the coverslip was inverted onto a 50-μl droplet of probe mixture on a parafilm-coated petri dish. The probe mixture comprised approximately 0.5 nM of each encoding probe for the MERFISH imaging, approximately 5 nM of each encoding probe for the two sequential rounds of two-colour FISH imaging, and 1 μM of a polyA-anchor probe (IDT) in 2× SSC with 30% v/v formamide, 0.1% wt/v yeast tRNA (approximately, Life Technologies) and 10% v/v dextran sulfate (D8906, Sigma). We then incubated the sample at 37 °C for 36–48 h. The polyA-anchor probe (/5Acryd/ TTGAGTGGATGGAGT GTAATT + TT + TT + TT + TT + TT + TT + TT + TT + T, where T+ is locked nucleic acid, and /5Acryd/ is 5' acrydite modification) was hybridized to the polyA sequence on the polyadenylated mRNAs and allowed these RNAs to be anchored to a polyacrylamide gel as described below. After hybridization, the samples were washed in encoding-probe wash buffer for 30 min at 47 °C for a total of two times to remove excess encoding probes and polyA-anchor probes. All tissue samples were cleared to remove fluorescence background as previously described[4,63]. In brief, the samples were embedded in a thin polyacrylamide gel and were then treated with a digestion buffer of 2% v/v sodium dodecyl sulfate (SDS; AM9823, Thermo Fisher), 0.5% v/v Triton X-100 (X100, Sigma) and 1% v/v proteinase K (P8107S, New England Biolabs) in 2× SSC for 36–48 h at 37 °C. After digestion, the coverslips were washed in 2× SSC for 30 min for a total of four washes and then stored at 4 °C in 2× SSC supplemented with 1:100 Murine RNase inhibitor (M0314S, New England Biolabs) for no longer than 2 weeks before imaging.

## MERFISH imaging

We used home-built imaging platforms for MERFISH imaging in this study, as previously described[64]. A commercial flow chamber (FCS2, Bioptechs) with a 0.75-mm-thick flow gasket (DIE F18524; 1907-100, Bioptechs) was used, and imaging buffer comprising 5 mM 3,4-dihydroxybenzoic acid (P5630, Sigma), 50 μM trolox quinone, 1:500 recombinant protocatechuate 3,4-dioxygenase (rPCO; OYC Americas),

1:500 Murine RNase inhibitor and 5 mM NaOH (to adjust pH to 8.0) in 2× SSC was used for all experiments. For sagittal slices, whole-tissue slices were imaged; for coronal slices, we imaged one hemisphere plus a narrow region near the midline in the other hemisphere. Two imaging schemes were used for the two different readout strategies:

(1) For the direct readout strategy, we first stained the sample with a readout hybridization mixture containing the readout probes associated with the first round of imaging, as well as a probe complementary to the polyA-anchor probe and conjugated via a disulfide bond to the dye Alexa488 at a concentration of 3 nM for imaging total polyadenylated mRNA. The readout hybridization mixture was composed of the readout-probe wash buffer containing 2× SSC, 10% v/v ethylene carbonate (E26258, Sigma) and 0.1 % v/v Triton X-100, supplemented with 3 nM each of the appropriate readout probes. The sample was incubated in this mixture for 15 min at room temperature and then washed in the readout-probe wash buffer supplemented with 1 µg ml$^{-1}$ DAPI for 10 min to stain nuclei within the sample. The sample was then washed briefly in 2× SSC and was ready for imaging. After the first round of imaging, the dyes were removed by flowing 2.5 ml of cleavage buffer comprising 2× SSC and 50 mM of Tris (2-carboxyethyl) phosphine (646547, Sigma) with 15 min incubation in the flow chamber to cleave the dyes linked to the readout probes through disulfide bond. The sample was then washed by flowing 1.5 ml 2× SSC. To perform the second round of imaging, we flowed 3.5 ml of the readout-probe mixture containing the appropriate readout probes across the chamber and incubated the sample in this mixture for 15 min. Then, the sample was washed by 1.5 ml of readout-probe wash buffer and 1.5 ml of imaging buffer was introduced into the chamber.

(2) For the two-step adaptor readout strategy, we first stained the sample with an adaptor probe hybridization mixture containing the adaptor probes associated with the first round of imaging. The readout hybridization mixture was composed of the readout-probe wash buffer containing 2× SSC and 30% v/v formamide (AM9342, Ambion), supplemented with 100 nM each of the appropriate adaptor probes. The sample was incubated in this mixture for 15 min at room temperature, washed in the readout-probe wash buffer and stained with a readout hybridization mixture containing 10 nM each of the two readout probes, as well as the polyA-anchor probe (Alexa488) at a concentration of 3 nM in the readout-probe wash buffer (2× SSC and 30% v/v formamide). The sample was incubated in this mixture for 15 min at room temperature, washed again and then washed in 2× SSC supplemented with 1 µg ml$^{-1}$ DAPI for 10 min to stain nuclei. Last, the sample was washed briefly in 2× SSC and was ready for imaging. After the first round of imaging, the dyes were removed by flowing 2.5 ml of cleavage buffer comprising 2× SSC, 30% formamide and 50 mM Tris (2-carboxyethyl) phosphine, supplemented with unlabelled common readout probes at 100 nM each to block unoccupied readout sequences on the adaptor probes to prevent crosstalk between rounds of hybridizations. The sample was incubated in this cleavage buffer for 15 min in the flow chamber, then washed by flowing 1.5 ml of readout-probe wash buffer. To perform the second round of imaging, we flowed 3.5 ml of the adaptor probe mixture containing the appropriate adaptor probes across the chamber and incubated the sample in this mixture for 15 min, washed by 1.5 ml of readout-probe wash buffer, and flowed 3.5 ml of the readout-probe mixture containing the common readout probes across the chamber and incubated the sample in this mixture for another 15 min. Then, the sample was washed again by 1.5 ml of readout-probe wash buffer and then 1.5 ml of imaging buffer was introduced into the chamber.

In the first round of imaging, we collected images in the 750-nm, 650-nm, 560-nm, 488-nm and 405-nm channels to image the first two readout probes (conjugated to Alexa750 and Cy5/Alexa647, respectively), the orange fiducial beads, the total polyA-mRNA signal by the polyA-anchor readout probe (Alexa488) and the nucleus signal by DAPI (405-nm channel). The latter two channels were used for cell segmentation as described below. For the second and all following imaging rounds, we collected images in the 750-nm, 650-nm and 560-nm channels for the two readout probes and fiducial beads. During each imaging round, for the fiducial beads, we took a single image at one z position for each field of view (FOV) on the surface of the coverslip using the 560-nm illumination channel as a spatial reference to correct for slight drift of the stage position over the course of imaging rounds. For imaging readout probes in the MERFISH rounds, we imaged multiple z positions in each FOV: for animal 2, we collected three or six 1.5-µm-thick z stacks; for all other animals, we collected five 1.5-µm-thick z stacks. We repeated the hybridization, wash, imaging and cleavage for all rounds to complete the 16 rounds of imaging for 32-bit MERFISH experiments. We then performed two additional rounds of two-colour FISH imaging to image the four additional genes, and these images were only acquired from one z plane per FOV. All buffers and readout probe mixtures were loaded with a home-built, automated fluidics system composed of three, 12-port valves (EZ1213-820-4, IDEX) and a peristaltic pump (MP3, Gilson).

## MERFISH image analysis and cell segmentation

All MERFISH image analysis was performed using MERlin (available at https://github.com/ZhuangLab/MERlin)[65], as previously described[64]. First, we identified the locations of the fiducial beads in each FOV in each round of imaging and used these locations to determine the $x–y$ drift in the stage position relative to the first round of imaging and to align images for each FOV across all imaging rounds. We then high-pass filtered the MERFISH image stacks for each FOV to remove background, deconvolved them using ten rounds of Lucy–Richardson deconvolution to tighten RNA spots, and low-pass filtered them to account for small movements in the centroid of RNAs between imaging rounds. Individual RNA molecules imaged by MERFISH were identified by our previously published pixel-based decoding algorithm using MERlin. After assigning barcodes to each pixel independently, we aggregated adjacent pixels that were assigned with the same barcodes into putative RNA molecules, and then filtered the list of putative RNA molecules to enrich for correctly identified transcripts as previously described for a gross barcode misidentification rate at 5% using MERlin.

We performed cell segmentation using the DAPI and total polyA-mRNA signals and a deep learning-based cell segmentation algorithm (Cellpose 2.0)[66,67]. We selected approximately 100 FOVs from the whole MERFISH dataset as the training images. To ensure the training set included images with different cellular densities and cytoarchitectural features, we included images from all different major brain regions for the training set generation. To train the human-in-the-loop Cellpose model, we used the DAPI and polyA images of these FOVs and first applied the 'cyto2' model in Cellpose with a diameter parameter of 100 pixels to segment the cells, followed by manually correcting cells that were mis-segmented and adding the cells that were missed by the automated cyto2 method. These human-curated images of approximately 100 FOVs were saved to form the training set and used to train the Cellpose model, and the trained model was used in the cell segmentation of all MERFISH data. Cells were segmented for each individual z plane, the centroid positions of the cells were determined in each z plane, and the centroids within distance of 2 µm in the $xy$ direction across different z planes were considered to be the same cell and were connected.

We assigned unique IDs for each segmented cell and assigned individual RNAs to segmentation boundaries of the cells based on whether they fell within those boundaries to obtain the cell × gene matrix, that is, the copy number of RNAs for each gene in each cell. The total number of segmented cells was about 10 million.

For the two sequential rounds of two-colour FISH imaging, we quantified the signal from these images by summing the fluorescence intensity of all pixels that fell within the segmentation boundaries of the cells associated with the imaged $z$ plane and normalized the signal by the areas of the cells in the $z$ plane.

## Preprocessing of MERFISH data

With the cell × gene matrix obtained as described above, we preprocessed the matrix by several steps: (1) the segmentation approach that we used generated a small fraction of putative cells with very small total volumes due to spurious segmentation artefacts, as well as some cells that overlapped in the $z$ dimension and were not properly separated. Thus, we removed the cells that had a volume of less than 50 μm³ or more than 1,500 μm³ for the 3-$z$ plane measurements, the cells that had a volume of less than 80 μm³ or more than 2,500 μm³ for the 5-$z$ plane measurements, and cells that had a volume of less than 100 μm³ or more than 3,000 μm³ for the 6-$z$ plane measurements. (2) To remove the differences in RNA counts due to different soma volumes captured in the images, we normalized the RNA counts per cell by the imaged volume of each cell. (3) We normalized the mean total RNA counts per cell to a same mean value (250 in this case) for each experiment. (4) We removed the cells that had total RNA counts in the top and bottom 1% quantile. (5) We removed potential doublets using Scrublet[68] as previously described. The cells with a doublet score higher than 0.25 were removed as doublets, which accounted for approximately 4% of the total cell number. After these preprocessing steps, approximately 9.3 million cells were kept for subsequent analysis.

## Integration of MERFISH data with scRNA-seq data

We grouped MERFISH data from all four animals for integration with scRNA-seq data. Hence, only the overlapping 1,122 genes between the two MERFISH gene panels used for all four animals were included in the cell × gene matrix for integration of MERFISH and scRNA-seq data and subsequent analyses.

We used the SeuratIntegration class from the ALLCools Python package[19,69] to integrate the MERFISH dataset and the scRNA-seq dataset. The integration works by co-embedding the two datasets in a common space and finding pairs of cells from the two datasets that are close to each other in the co-embedded space. The identified close pairs are termed anchors, which were used for transferring cell-type labels and imputing gene expressions from the scRNA-seq dataset to the MERFISH dataset. We performed co-embedding of the two datasets by a canonical correlation analysis (CCA)-based integration algorithm[19,69]. To integrate more than 10 million cells from the two datasets while achieving a fine resolution for more than 5,000 transcriptionally distinct cell clusters identified in the scRNA-seq data, we performed two rounds of integration.

First, we divided the cells from both datasets into 50 integration partitions. We used the scRNA-seq dataset to define the partitions. Each integration partition was a group of subclasses that were close in the transcription space. We subset the genes in the scRNA-seq dataset to the genes measured by MERFISH. Then, we preprocessed the dataset using the Scanpy pipeline[70]: normalized the total count of each cell to 1,000, log1p transformed the counts and scaled the transformed counts to $Z$ scores. We reduced the dimensionality to 100 principal component analysis (PCA) dimensions and calculated the 15 nearest neighbours of each cell in the PCA space. From the nearest neighbour graph, we calculated a connectivity graph of subclasses where each node was a subclass and the weight of each edge was the number of edges in the nearest neighbour graph that connected cells from the two subclasses. Then, we used the direct k-way cuts method from the METIS graph partitioning library[71] to divide the 338 subclasses into 50 integration partitions. This method aimed to evenly distribute cells into partitions while minimizing the sum weight of cut edges.

In the first round of integration, we transferred the integration-partition labels from the cells in the scRNA-seq dataset to the cells in the MERFISH dataset. We subset the genes in the scRNA-seq dataset to the genes measured by MERFISH. Then, we independently preprocessed the scRNA-seq and MERFISH datasets by the Scanpy pipeline[70]: normalized the total count of each cell to 1,000, log1p transformed the counts and scaled the transformed counts to $Z$ scores. We combined the two datasets and performed PCA to reduce the dimensionality to 100. We ran CCA to co-embed the scRNA-seq cells and MERFISH cells into a 100-dimensional space. To co-embed the large number of cells from the two datasets, the CCA was first performed on randomly downsampled scRNA-seq and MERFISH datasets, each containing 100,000 cells. Then, the CCA coordinates of the full datasets were calculated by a linear transformation from the gene expression space to the CCA space. We found the five nearest neighbours across the two datasets in the CCA space. We defined all pairs of cells from the two datasets that were mutual nearest neighbours as integration anchors. Then, we used the label_transfer function from the SeuratIntegration class to transfer the integration-partition labels from the scRNA-seq dataset to the MERFISH dataset. For each MERFISH cell, the label_transfer function calculated the probability of assigning the MERFISH cell to every integration partition based on the 100 nearest-neighbour anchor cells from the scRNA-seq dataset in the PCA space. We set the integration-partition label of a MERFISH cell to be the one with the highest probability (that is, the integration partition that had the highest fraction of cells in the 100 nearest-neighbour anchor cells) and defined this probability as the confidence score of the transferred partition label.

In the second round of integration, we transferred subclass and cluster labels from the scRNA-seq dataset to the MERFISH dataset. We performed this round of integration for each integration partition separately. We subset the genes in the scRNA-seq dataset to the genes measured by MERFISH, normalized the total count of each cell to 1,000 and log1p transformed the counts. We used the genes that were highly variable in each integration partition. To this end, we calculated the dispersions of all the selected genes using the highly_variable_genes function from the Scanpy package[70]. Only genes with log dispersions greater than zero were kept for integration. Using the same method for the first round of integration, we transferred the subclass and cluster labels from the scRNA-seq dataset to the MERFISH dataset and calculated the confidence scores for label transfer. Because a cell-type label is transferred correctly to a cell only when both the integration-partition label and the cell-type label within the integration partition were transferred correctly, we adjusted the confidence scores of the subclass and cluster label transfer by multiplying them with the integration-partition label-transfer confidence scores. Among the 9.3 million cells that were integrated with the scRNA-seq data, we further removed the cells that substantially passed the midline in the coronal slices and those that passed the posterior edge of the CCF in the sagittal slices, as well as six fractured tissue slices (see 'MERFISH image registration to the CCF' for details on CCF registration); 8.4 millions cells remained after this filtering step. The cell-by-gene matrices of the remaining 8.4 millions cells can be downloaded from both the Allen Brain Cell Atlas and the CELLXGENE database, and are displayed on the CELLXGENE database (see 'Data availability' section). In addition, we further filtered the cells by the label transfer confidence scores, and the 5.8 million cells that passed the thresholds for the subclass and cluster label transfer confidence scores are included in the cell metadata file that can be downloaded from and are displayed on the Allen Brain Cell Atlas (see 'Data availability' section).

## Imputation of transcriptome-wide gene expressions of individual cells in MERFISH images

On the basis of the integration of MERFISH and scRNA-seq data, we also imputed the transcriptome-wide gene expression for each cell in the MERFISH images using the method previously described[69]. In short,

the imputed expression profile of a MERFISH cell was calculated as the weighted average of the expression profiles of its 30 nearest-neighbour anchor cells in the scRNA-seq dataset in the co-embedded PCA space. The weights were based on the distance between the scRNA-seq cells to the MERFISH cell and were calculated by the find_nearest_anchor function from the SeuratIntegration class using default parameters.

We evaluated the validity of the imputation results by comparing them with the gene expression measured by MERFISH and with the previously measured spatial expression patterns in Allen Brain Atlas in situ hybridization data[20] for the genes included in the MERFISH gene panel, and with the Allen Brain Atlas in situ hybridization data only for the genes not included in the MERFISH gene panel. We performed two correlation analysis for comparing imputation results with the MERFISH measurement results. First, we calculated the mean expression level in every cluster from the imputation results and the MERFISH measurement results for each gene. We then quantified the Pearson correlation coefficient between the imputed cluster means and MERFISH-measured cluster means across all clusters for each gene. Second, we calculated the mean expression levels of every imaged FOVs from the imputation results and the MERFISH-measurement results for each gene, and then quantified the Pearson correlation coefficient between the imputed FOV means and MERFISH-measured FOV means across all imaged FOVs for each gene. The first comparison evaluated how well the relative expression levels of genes in different clusters were recapitulated by the imputation and the second comparison evaluated how well the spatial variation in gene expression was recapitulated by the imputation.

For the genes not included in the MERFISH, we visually compared the spatial patterns of gene expression determined by imputation with those determined in Allen Brain Atlas in situ hybridization data.

## MERFISH image registration to the CCF

Registration of MERFISH data to the Allen Mouse Brain CCFv3 was performed in a two-step process involving the reconstruction of 2D MERFISH tissue slices to a 3D volumetric image through alignment of DAPI signals in the MERFISH images to the Nissl template images in the Allen Reference Atlas, followed by a 3D refinement using landmarks based on cell types with known localizations in the CCF. For the initial reconstruction, we used the DAPI channel in the MERFISH images of individual brain slices and the Nissl template images in the Allen Reference Atlas, which is aligned to the Allen CCF. For each MERFISH sample from the same animal, brain slices were ordered and rotated to match coronal or sagittal orientation of the CCF. Coronal slices were cropped approximately 200 μm past the midline, whereas sagittal slices were cropped at the posterior end of the cerebellum. In each animal, key slices containing recognizable landmarks were used to identify corresponding CCF planes, and all remaining CCF planes were determined by linear interpolation. To aid the registration process, features in the DAPI image were enhanced by highlighting pixels containing cell types that localized to known brain regions (for example, VLMCs at the brain surface, ependymal cells in the ventricles, granular cells in the dentate gyrus, among others). The corresponding features in the Nissl image were also highlighted using the CCF annotations and/or morphological operations. Finally, each DAPI–Nissl image pair was registered with an affine and then B-spline transformation using the program Elastix[72]. Each transformation was then applied to the cell positions to find their initial position in the CCF space.

In the second alignment step to refine the CCF registration, an additional 3D–3D registration was performed using additional selected cell types from the MERFISH data that are known to be localized to certain brain regions in the CCF. In total, 36 suitable cell types were identified along with their corresponding brain region annotations in the CCF, as well as two level 1 space modules (SM_CTX and SM_RSP) that delineated the cells in the isocortex. These selected cell types (or spatial modules) were each randomly assigned an intensity label,

and a 3D volumetric image was generated using their initial positions in the CCF space from the first reconstruction step. A second target 3D image was generated but using only the CCF annotations; for each selected cell type, the corresponding brain region annotations in the CCF were assigned the intensity label for that cell type or spatial module, and all other annotated regions were removed. As before, for certain cell types, morphological operations on certain annotations were used to denote the midline, tissue surface or hollow ventricles. Finally, these two 3D images were registered using a B-spline transformation and the cell positions were refined.

After the MERFISH data were registered to the CCF, each MERFISH cell was assigned a 3D coordinate (ccfx, ccfy and ccfz), indicating its spatial location in the CCF space, where ccfx indicates the coordinate value along the rostral–caudal direction, ccfy indicates the coordinate value along the dorsal–ventral direction and ccfz indicates the coordinate value along the lateral–medial direction. Each MERFISH cell was also assigned a brain region annotation ID as defined in the CCF, indicating its brain region identity.

For visualization in individual figures, we presented the MERFISH-imaged cells in the experimental coordinates, but reverse transformed the brain region boundaries defined in the CCF into the experimental coordinates by reversing the above-described MERFISH image-to-CCF transformation.

As a cautionary note, although our CCF registration of the MERFISH-derived cell atlas allows characterization of cell-type composition and organization in different brain regions, alignment errors could exist in CCF registration due to the differences between individual mouse brains and the average template represented by the Allen CCFv3, as well as the deformation of tissue sections that were not completely corrected for during image alignment. Improvement in CCF-registration accuracy is an active research topic and the CCF reference itself is also actively evolving. Thus, our current CCF registration provides a starting point, and future method development in this area will help improve the accuracy of CCF registration.

## Neurotransmitter identities of the neurons

We assigned neurotransmitter identity to the neurons based on their expression of canonical neurotransmitter transporter genes. Specifically, *Slc17a7*, *Slc17a6* and *Slc17a8* were used for identifying glutamatergic neurons, *Slc32a1* for GABAergic neurons, *Slc6a4* for serotonergic neurons, *Slc6a3* for dopaminergic neurons, *Slc18a3* for cholinergic neurons, *Slc6a5* for glycinergic neurons and *Slc6a2* for noradrenergic neurons. In addition, *Hdc*, which is involved in histamine synthesis, was used to mark the histaminergic neurons. For all of these genes, we used an expression threshold of RNA counts per cell $n \geq 2$, determined by MERFISH, to assign neurotransmitter identity to individual neurons.

## Spatial module analysis

We did two rounds of spatial module analysis to delineate molecularly defined brain regions based on local cell-type composition.

For the first round of spatial module analysis, we defined a local cell-type-composition vector for each cell to characterize its neighbourhood composition of cell types at the subclass level. We began by finding the 50 spatially nearest neighbours for each cell using scikit-learn[73]. Because vascular and immune cells are usually randomly distributed across most brain regions, we excluded them from the spatial module analysis. Then, we assigned a weight to each neighbour cell $j$ of a cell $i$ as:

$$\text{Weight}_{i,j} = \exp(-(D_{i,j}/D_i^{(0)})^2)$$

Where $D_{i,j}$ is the spatial distance between cell $i$ and cell $j$, and $D_i^{(0)}$ is the distance scaling factor. Because different brain regions have different cell densities, we let $D_i^{(0)}$ be adjustable based on the local cell density and defined $D_i^{(0)}$ as two times the distance between cell $i$ and its fifth

nearest spatial neighbour. Then, we defined the local cell-type-composition vector of a cell from its neighbour cell types and weights. Each element of a local cell-type-composition vector corresponds to a cell type, and the value is the sum of the weights of the spatial neighbours that belong to this cell type.

We generated the first level of spatial modules by clustering cells based on their local cell-type-composition vectors at the subclass level. We normalized the local cell-type-composition vectors by their L2 norms and ran the Leiden clustering method to cluster the cells. We manually curated the clusters by merging the clusters that did not form clear spatial boundaries and annotated the clusters based on the major brain regions that they corresponded to. This round of analysis gave level 1 spatial modules.

We then generated the level 2 spatial modules for each level 1 spatial module separately. Because the spatial heterogeneity of cell types within individual major brain regions are mainly due to neurons, we only considered neurons for the second round of spatial module analysis. We calculated the local cell-type-composition vectors using the same method described for the first round of spatial module analysis with two modifications. The first modification was that we considered both subclasses and clusters to define the local cell-type-composition vectors – the subclass-based vector was concatenated with the cluster-based vector to form the overall vector. The second modification was that we used a shorter distance scaling factor $D^{(0)}_i$ for the higher spatial resolution in this round. We defined $D^{(0)}_i$ as the distance between cell $i$ and its fifth nearest spatial neighbour. Then, we used the same method described for the first round of spatial module analysis to cluster cells based on their local cell-type-composition vectors to generate level 2 spatial modules.

## Spatial gradient analysis

All cells with subclass label-transfer confidence scores greater than 0.8 were used in the spatial gradient analysis. To define the degree of how discrete or how well separated individual clusters were within each subclass, for each cell, we calculated its 'neighbourhood purity' defined by the fraction of cells that had the same cell-cluster label as the centre cell among its 50 nearest neighbours in the gene expression space. The discreteness of a cell cluster was defined by the mean value of the neighbourhood purities of all cells within the cluster. We then determined the median cluster discreteness of a subclass as a measure of how discrete individual clusters were within the subclass.

To visualize the spatial gradient of the subclasses or groups of transcriptionally similar subclasses, PCA was used to reduce dimensionality of the normalized expression data and to calculate a 'pseudotime' value for each cell as previously described[8]. Next, spatial gradients were visualized by representing gene expression profiles of the cells using either PC1 or the pseudotime value of individual cells on the spatial maps. In addition, correlation of the PC1 or pseudotime values and the spatial coordinate of the cells were plotted. For the IT neurons in the isocortex, cortical depth was used as the spatial coordinate and was calculated for individual neurons as previously described[8] for coronal slices in the region between Bregma approximately −0.8 and approximately +1.7 where the layer 6b CTX cells formed a clear thin layer at the bottom border of the isocortex. For the D1 and D2 medium spiny neurons, locations along the dorsolateral–ventromedial axis were used as spatial coordinate values and were calculated using the ccfy (dorsal–ventral) and ccfz (medial–lateral) locations of individual cells. For lateral septal complex neurons and tanycytes, locations along the dorsal–lateral axis (ccfy) were used as spatial coordinate values.

## Cell–cell interaction analysis

We performed cell–cell interaction analysis at the subclass level. All cells with a subclass label-transfer confidence score greater than 0.8 were used in this analysis. We divided cells into major brain regions based on their CCF coordinates. Owing to the high complexity of cell-type

compositions of the hypothalamus, midbrain and hindbrain, we further divided these regions each into two regions: the hypothalamus was divided into the anterior and posterior hypothalamus; the midbrain was divided into the anterior and posterior midbrain; and the hindbrain was divided into the pons and the medulla. For the hypothalamus and midbrain, the region was divided based on the cell locations along the rostral–caudal axis (ccfx), specifically, the mean value of the minimum and maximum ccfx value for all the cells within the region was used to divide the region into the anterior and posterior parts. We only considered the subclasses that were either enriched or had a sufficient abundance in each brain region for the cell–cell interaction analysis. For neuronal subclasses, we used the enrichment score as described in the caption for Fig. 2b. For the anterior hypothalamus, posterior hypothalamus, anterior midbrain, posterior midbrain, pons and medulla, we used an enrichment score threshold of 6 to stringently select cells in these regions. For the other brain regions, we set the enrichment score threshold to 2. For astrocytes, we used an enrichment score threshold of 1 for all brain regions. For the remaining subclasses of non-neuronal cells, we considered them in a brain region if the total cell number of that subclass was greater than 50 in this region.

For each subclass pair within each region, we determined the number of cell pairs (one from each subclass) that were in contact or proximity and compared the number of contact or proximal cell pairs with a null distribution generated by randomly shifting spatial positions of the cells locally[11]. Two cells were considered in contact or proximity if the distance between the cell centroid positions was within a distance threshold ($R_{proximal}$). We first defined $R_{proximal}$ to be 15 μm, which is comparable to the soma size of the cells in the mouse brain. To generate the null distribution by randomly shifting spatial positions of the cells locally, for each round of randomization, we shifted the spatial location of each cell to a random position within 100 μm from its original location. We performed 1,000 rounds of randomization. After each round, we calculated the number of cell pairs that were in contact or in proximity between every pair of subclasses. For each pair of subclasses, we fitted the distribution of the number of contact/proximal cell pairs generated by 1,000 randomizations to a normal distribution to generate the null distribution. We then compared the observed contact/proximal cell pair number with the null distribution to determine the enrichment fold change and the $P$ value of the enrichment. Then, we used the Benjamini–Hochberg multiple-hypothesis testing correction method to adjust the $P$ values. We used the adjusted $P$ value threshold of 0.05 and the number of observed proximal pair threshold of 50 to select the pair of subclasses that showed significant probability to be in contact or in proximity and called these subclass pairs as interacting cell-type pairs.

As the stringent distance threshold, $R_{proximal} = 15$ μm, may eliminate some cell-type pairs that communicate through paracrine signalling, we also relaxed this distance threshold to a greater value ($R_{proximal} = 30$ μm), but for cell-type pairs identified with this relaxed distance threshold, we further required that at least one ligand–receptor pair was upregulated in the proximal cell pairs compared with non-proximal cell pairs (see below) to call these cell types as interacting cell-type pairs.

## Ligand–receptor analysis and analysis of other genes upregulating in interacting cell pairs

We performed the ligand–receptor analysis at the subclass level. All cells with a subclass label-transfer confidence score greater than 0.8 were used in this analysis. We used the CellChat database[74] to define the ligand–receptor pairs. For a ligand–receptor pair $k$, we defined the ligand–receptor expression score for a pair of cells $i$ and $j$ as:

$$S_{k,i,j} = \log(1 + \Pi_{p,q} L_{k,i,p} * R_{k,j,q})$$

Where $L_{k,i,p}$ is the expression level of the $p$-th component of the ligand of the ligand–receptor pair $k$ in the cell $i$; $R_{k,j,q}$ is the expression level of

the $q$-th component of the receptor of the ligand–receptor pair $k$ in the cell $j$. The expression levels used here were the imputed gene expression results as described in the section 'Imputation of transcriptome-wide gene expressions of individual cells in MERFISH images'.

We performed ligand–receptor pair analysis for the cell-type pairs that showed statistically significant proximity compared with the null distribution as described in the previous section 'Cell–cell interaction analysis', using $R_{proximal} = 30\,\mu m$. For a pair of cell types and a ligand–receptor pair, we calculated the distributions of ligand–receptor expression scores for all proximal cell pairs, that is, cell pairs with a soma centroid distance smaller than $R_{proximal}$, from this cell-type pair (one cell from each cell type). Then, we randomly selected the same number of cell pairs from this cell-type pair with a soma centroid distance greater than $R_{proximal}$. We calculated the distributions of ligand–receptor expression scores for the non-proximal cell pairs. We used one-sided Welch's $t$-test to test whether the mean ligand–receptor expression scores were significantly higher in proximal cell pairs than the scores in the non-proximal cell pairs. Then, we used the Benjamini–Hochberg multiple-hypothesis testing correction method to adjust the $P$ values. We selected significant ligand–receptor pairs that satisfied the following three criteria: the mean of ligand–receptor expression score was at least twofold higher in the proximal cell pairs than those in the non-proximal cell pairs; the adjusted $P$ value was less than 0.01; and the ligand–receptor expression scores were greater than zero in at least 40% of the proximal cell pairs. Using this approach, we determined the ligand–receptor pairs that were statistically significantly upregulated in the proximal cell pairs compared with the non-proximal cell pairs in each cell-type pair that showed statistically significant proximity using $R_{proximal} = 30\,\mu m$.

We then used a similar approach to determine other genes that were upregulated in the proximal cell pairs compared with non-proximal cell pairs in each cell-type pair. We first determined the highly variable genes for each cell type. Only highly variable genes were considered for this gene upregulation analysis. For each cell type A that showed significant proximity with another cell type B as compared with the null distribution, we divided the type A cells into two groups based on whether they were within $R_{proximal}$ of any type B cells. For each gene, we calculated the expression distributions in the two groups respectively and used one-sided Welch's $t$-test to test whether the mean expression was significantly higher in the first group than that in the second group. We used the Benjamini–Hochberg multiple-hypothesis testing correction method to adjust the $P$ values. We selected significantly upregulated genes using the following criteria: the mean expression level was at least twofold higher in the proximal cell pairs than those in the non-proximal cell pairs, and the adjusted $P$ values were less than 0.01.

## Statistics and reproducibility

Four replicate mice, one female and three males, were imaged under each condition. From the four replicate mice imaged for the identification and spatial mapping of cell types, a total of approximately 10 million cells were imaged and segmented, which generated a sufficient number of single-cell profiles and gave sufficient statistics for the effect sizes of interest. No statistical methods were used to predetermine sample size. The mice were randomly chosen. For each mouse, the imaging experiments were definitive, and no randomization was necessary for this study, hence the experiments were not randomized. The investigators were not blinded during experiments and outcome assessment because all images were taken under the same condition, and the results were quantitative, which did not require subjective judgement.

The sample sizes for the violin plots in Fig. 6g (from left to right) are 14, 13, 14 and 9 brain regions. The $P$ values in Fig. 6c, Extended Data Fig. 9 and Supplementary Table 4 were calculated by a one-sided permutation-based test described in the cell–cell interaction analysis section of the Methods. The displayed $P$ values were adjusted by the Benjamini–Hochberg multiple-hypothesis testing correction.

## Reporting summary

Further information on research design is available in the Nature Portfolio Reporting Summary linked to this article.

## Data availability

Raw and processed MERFISH data, as well as the MERFISH codebook and probes used in this work, can be accessed via the Brain Image Library[75]. Processed MERFISH data are also accessible and explorable in an interactive manner through two platforms: (1) the Allen Brain Cell Atlas (https://knowledge.brain-map.org/data/5C0201JSVE04WY6DMVC/ explore; https://alleninstitute.github.io/abc_atlas_access/descriptions/ Zhuang-ABCA-1.html; https://alleninstitute.github.io/abc_atlas_access/ descriptions/Zhuang-ABCA-2.html; https://alleninstitute.github.io/ abc_atlas_access/descriptions/Zhuang-ABCA-3.html; https://allenin-stitute.github.io/abc_atlas_access/descriptions/Zhuang-ABCA-4.html) and (2) the CELLxGENE database (https://cellxgene.cziscience.com/ collections/0cca8620-8dee-45d0-aef5-23f032a5cf09). The scRNA-seq datasets (FASTQ files) obtained by the Allen Institute are available at NeMO (https://assets.nemoarchive.org/dat-qg7n1b0). The processed scRNA-seq data along with the transcriptomic cell-type taxonomy were visualized at the Allen Brain Cell Atlas (mouse whole-brain cell-type atlas, https://portal.brain-map.org/atlases-and-data/bkp/abc-atlas). Instruction for access of the processed scRNA-seq data is available at https://github.com/AllenInstitute/abc_atlas_access/blob/main/ descriptions/WMB-10X.md. The CellChat database[74] is available at http://www.cellchat.org/. Source data are provided with this paper. Bulk RNA-seq data of the whole mouse brain are available at NCBI GEO data repository (GSE246919).

## Code availability

Code for the MERFISH image analysis is available at https://github.com/ ZhuangLab/MERlin and on Zenodo[65]. Additional code for data analysis is available at https://github.com/ZhuangLab/whole_mouse_brain_ MERFISH_atlas_scripts_2023 and on Zenodo[76]. Code for the MERFISH image acquisition is available at https://github.com/ZhuangLab and on Zenodo[77].

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

**Acknowledgements** We thank other Allen Institute team members for their contributions in generating the scRNA-seq data and transcriptomic cell-type taxonomy used as a reference in this study; and members of the Zhuang laboratory and the Allen Institute for helpful discussions. This work was supported in part by the US National Institutes of Health (BRAIN Initiative Cell Census Network (BICCN) grant U19MH114830 to X.Z. and H.Z.). X.P. is a Howard Hughes Medical Institute Jane Coffin Childs postdoctoral fellow. X.Z. is a Howard Hughes Medical Institute investigator.

**Author contributions** X.Z. conceived the project. M.Z., X.P., W.J., S.W.E., Z.Y., H.Z. and X.Z. designed the experiments. M.Z., X.P., W.J., A.R.H. and X.Z. designed the data analyses. Z.Y. designed the MERFISH gene panel with input from M.Z., X.P., W.J., H.Z. and X.Z. M.Z., W.J., S.W.E. and L.C. performed the MERFISH experiments and image decoding. M.Z., X.P., W.J., A.R.H. and Z.L. performed the data analysis. K.A.S., B.T., Z.Y. and H.Z. provided the scRNA-seq data and transcriptomic cell-type taxonomy. M.Z., X.P. and X.Z. wrote the paper with input from W.J., A.R.H., S.W.E., Z.L., L.C., K.A.S., B.T., Z.Y. and H.Z.

**Competing interests** X.Z. is a co-founder and consultant of Vizgen. H.Z. is on the scientific advisory board of MapLight Therapeutics, Inc.

**Additional information**
**Correspondence and requests for materials** should be addressed to Xiaowei Zhuang.

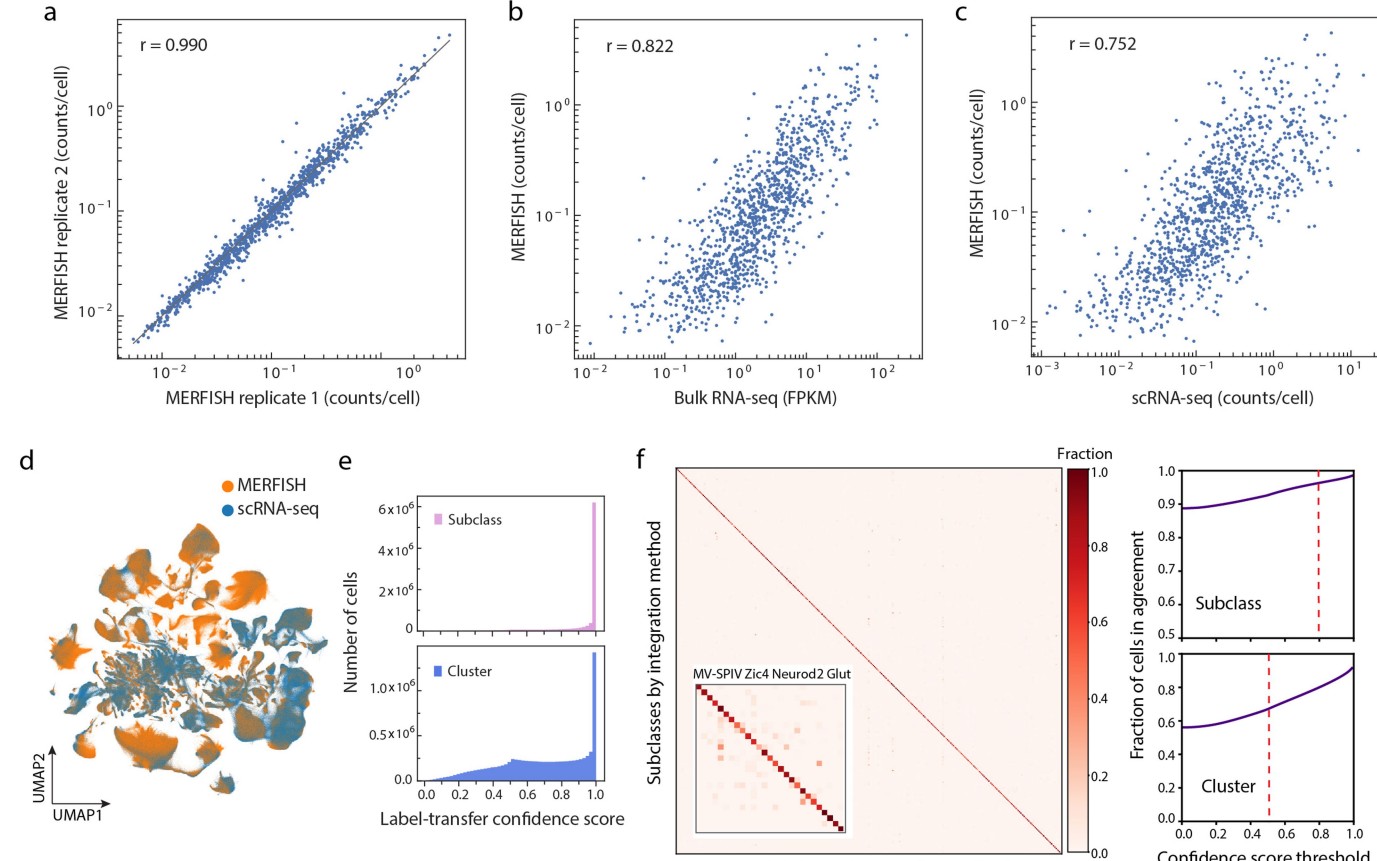

**Extended Data Fig. 1 | Correlation and integration of MERFISH data and RNA-seq data. a**, Correlation plot of the average copy number per cell of individual genes measured by MERFISH from two replicate animals. The black solid line indicates equality. The Pearson correlation coefficient is r = 0.990. **b**, Correlation plot of the average copy number per cell of individual genes determined by MERFISH versus the expression levels determined by bulk RNA-seq of whole mouse brain. The Pearson correlation coefficient is r = 0.822. **c**, Correlation plot of the average copy number per cell of individual genes determined by MERFISH versus those determined by scRNA-seq of whole mouse brain. The Pearson correlation coefficient is r = 0.752. **d**, UMAP of the integrated MERFISH and scRNA-seq data with all MERFISH and scRNA-seq cells displayed. Cells are coloured by experimental modalities. **e**, Distributions of confidence scores of subclass label transfer (top) and cluster label transfer

(bottom) for individual MERFISH cells. **f**, Left: Correspondence between the subclass classification of MERFISH cells determined by integration of MERFISH and scRNA-seq data (Integration method) and by identifying the scRNA-seq cluster with most similar transcriptional profile to the MERFISH cells (Mapping method). Confusion matrix shows the fraction of cells from any given subclass determined by the Integration method that was assigned to individual subclasses determined by the mapping method. Insets: Correspondence plots between the cluster classification of MERFISH cells determined by the two methods for an example subclass: MV-SPIV Zic4 Neurod2 Glut. Right: Fraction of cells showing classification agreement between the two methods as a function of the confidence score threshold at subclass level (top) and cluster level (bottom) used in the Integration method. Red dashed lines indicate the confidence score threshold used in this work.

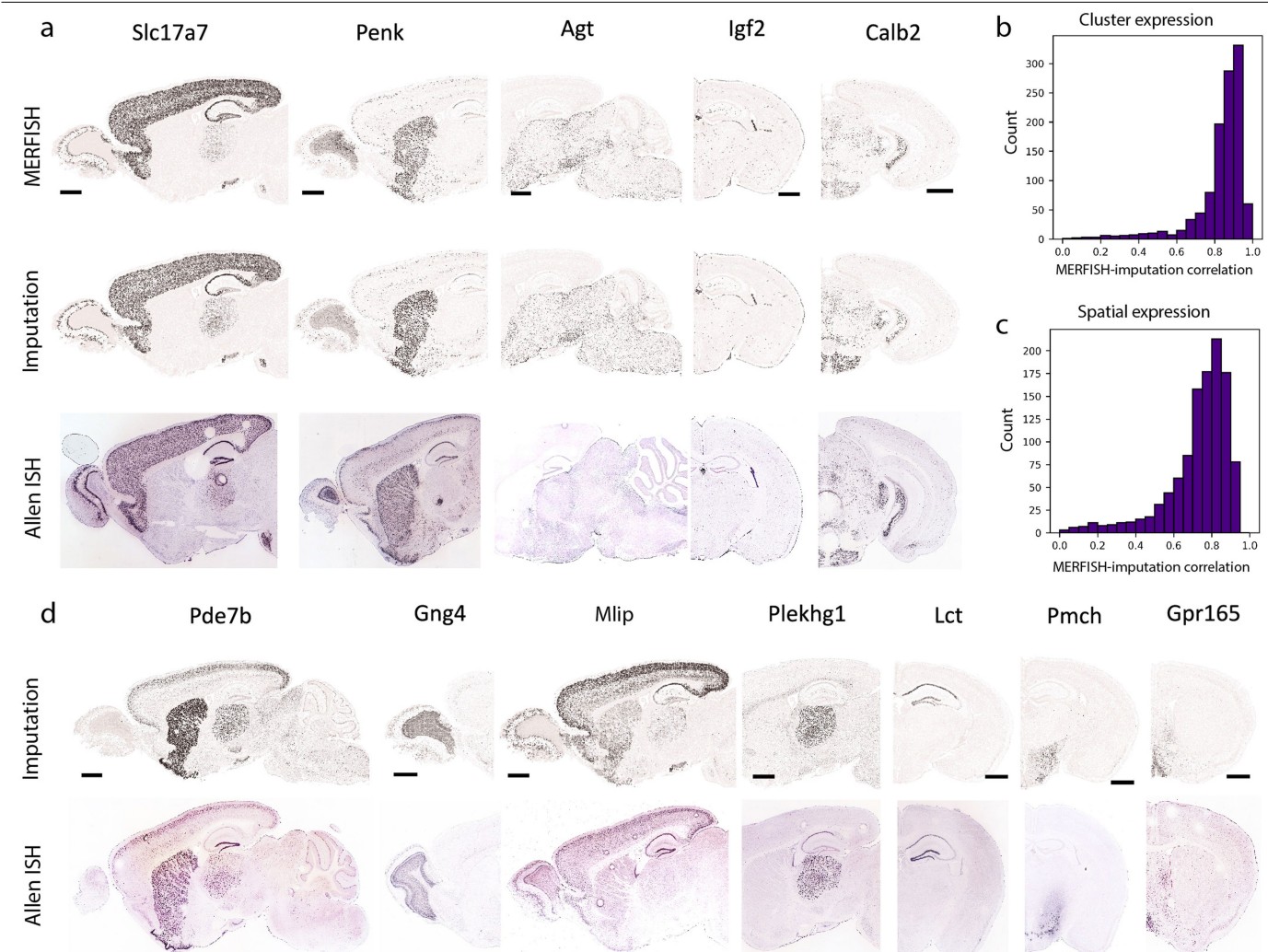

**Extended Data Fig. 2 | Comparison of gene-expression results imputed from MERFISH and scRNA-seq data integration with the MERFISH measurement results and Allen in situ hybridization data. a**, Examples of spatial gene-expression patterns from MERFISH measurement (top row), imputation results (middle row), and in situ hybridization data from the Allen brain atlas (bottom row). **b,c**, The distributions of Pearson correlation coefficients between MERFISH measurement results and imputation results. **b**, For each gene, a correlation coefficient was calculated for mean expression levels in individual cell clusters between MERFISH measurement results and

imputation results. **c**, For each gene, a correlation coefficient was calculated for mean expression levels of individual imaging fields of view (200 μm × 200 μm) between MERFISH measurement results and imputation results. Correlation-coefficient distributions across all genes in the MERFISH panel are shown. **d**, Examples of spatial gene expression patterns from imputation results (top row) and in situ hybridization data from the Allen brain atlas (bottom row). The genes shown in (**d**) were not measured by MERFISH. Scale bars in **a,d**: 1 mm. The Allen Brain Atlas in situ hybridization data in panels **a** and **d** are taken from https://mouse.brain-map.org/[20].

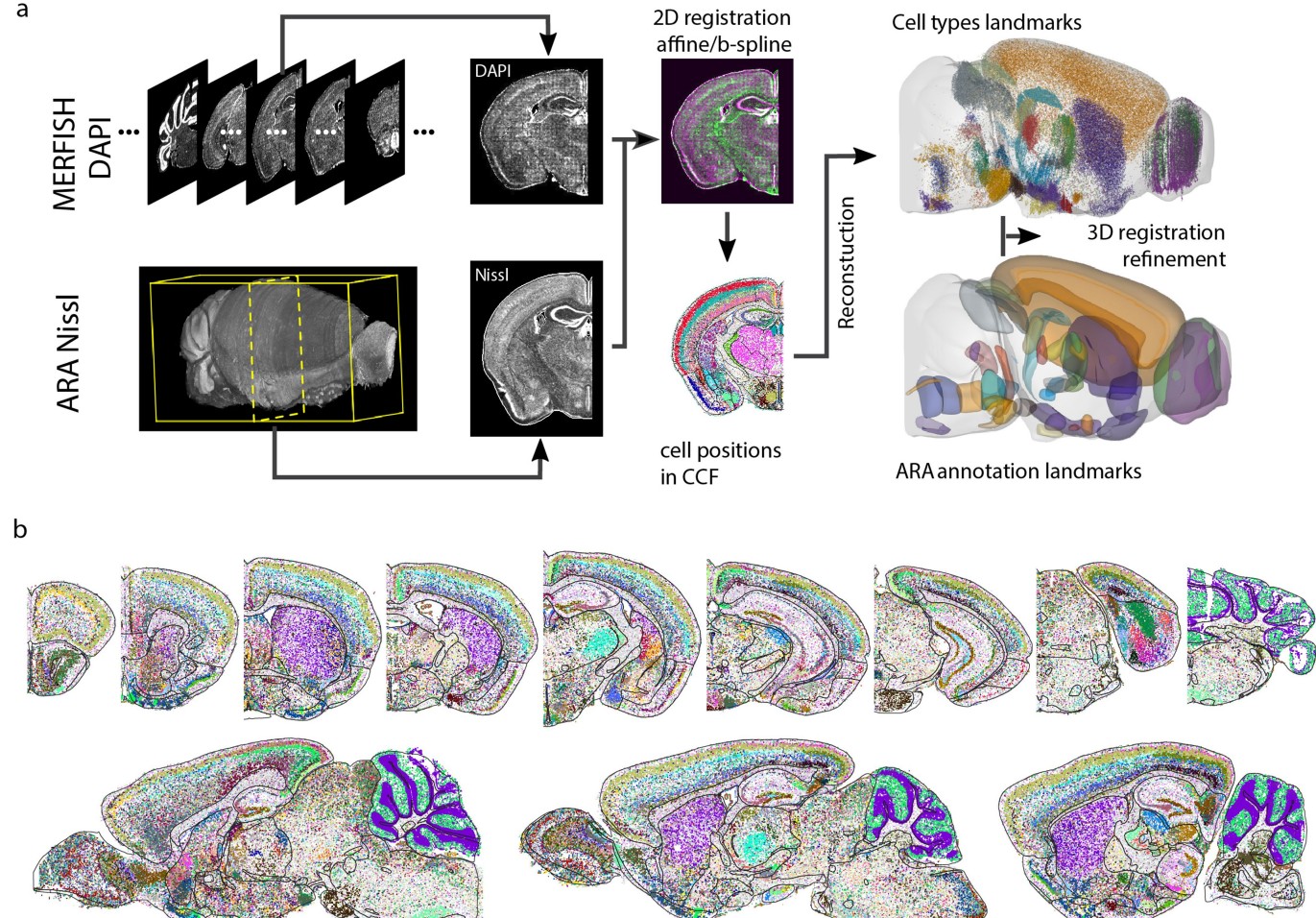

**Extended Data Fig. 3 | CCF registration of MERFISH images. a**, Workflow of CCF registration of the MERFISH images. MERFISH images were registered to the Allen Mouse Brain CCFv3[21] using a two-step procedure. First, DAPI images taken during MERFISH imaging were aligned to the Nissl template images in the Allen Reference Atlas (ARA, adapted from https://mouse.brain-map.org/static/atlas), which allowed an initial, coarse alignment of the MERFISH images to the Allen CCF. Second, cell-type with known locations in the CCF were selected as landmarks (e.g., layer-specific cortical neurons, neurons in the dente gyrus, etc.) and used to refine the CCF alignment (see Methods for details). The 3D brain images were generated using Brainrender[78]. **b**, Spatial maps of cells in the same coronal and sagittal sections as shown in Fig. 1c, but with cells coloured by their cluster identities. The underlying contour lines marking the brain region boundaries were generated using coordinates from the Allen Mouse Brain CCFv3 (ref. 21). Scale bar: 1 mm.

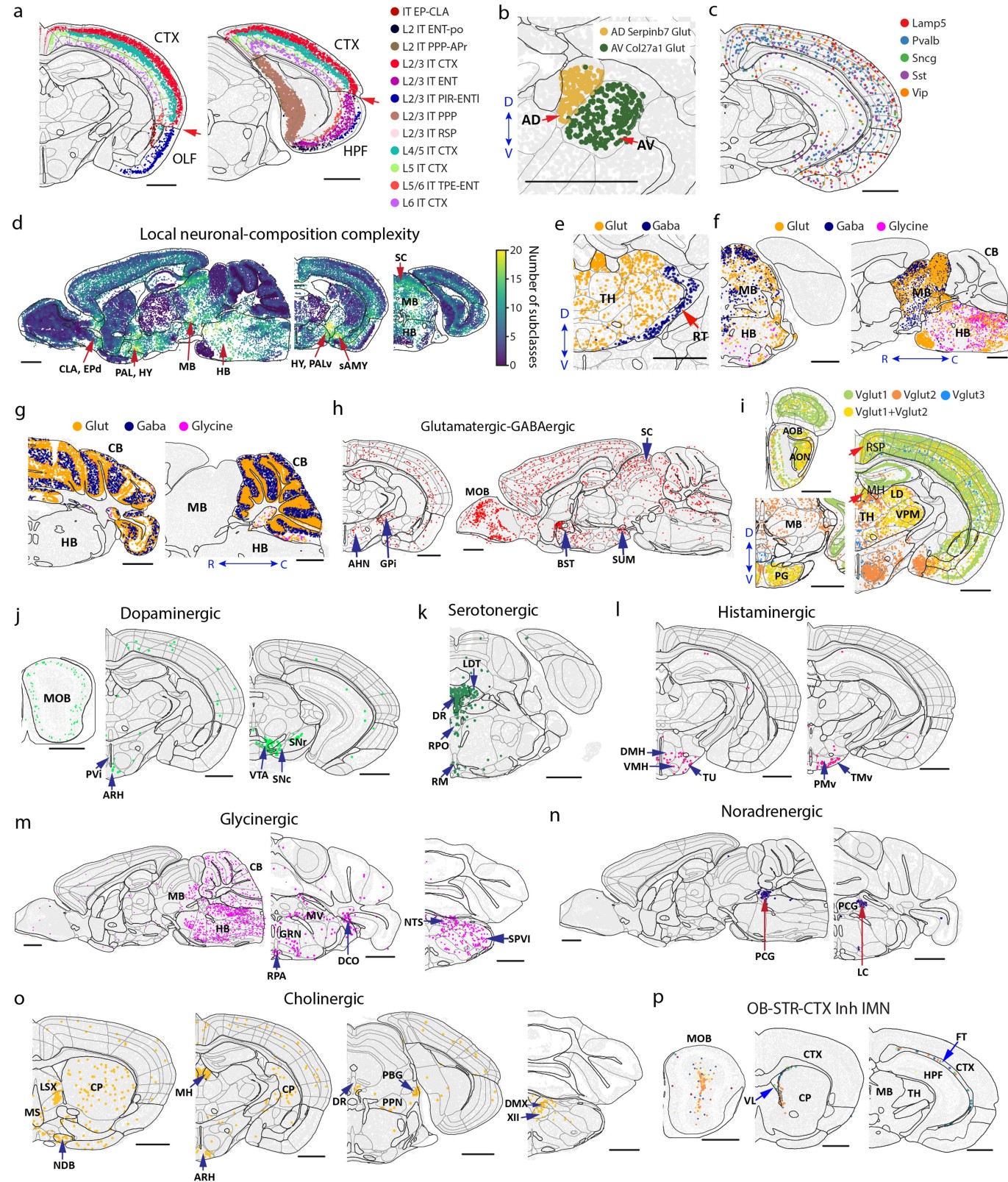

**Extended Data Fig. 4** | See next page for caption.

**Extended Data Fig. 4 | Spatial distributions of different neuronal cell types and neurotransmitter usage. a**, Spatial distributions of different IT subclasses showing the separation between IT neurons in the isocortex (CTX) and those in the olfactory areas (OLF, left) and in the hippocampal formation (HPF, right). Red arrows mark the boundaries between CTX and OLF and between CTX and HPF defined in the CCF. Cells are coloured by subclass identities. **b**, Spatial distributions of the two subclasses, AD Serpinb7 Glut and AV Col27a1 Glut, in the anterodorsal (AD) and anteroventral (AV) nucleus of the thalamus, respectively. **c**, Spatial distributions of five inhibitory neuronal subclasses, marked by *Lamp5*, *Pvalb*, *Sst*, *Vip*, and *Sncg*, across CTX, HPF, OLF and cortical subplate (CTXsp). **d**, Spatial heatmap of local neuronal-composition complexity. The local neuronal-composition complexity of any given cell is defined as the number of different neuronal cell types (at the subclass level) present in the 50 nearest-neighbour neurons surrounding that cell. PAL, Pallidum; PALv, Pallidum, ventral region; sAMY, Striatum-like amygdalar nuclei; SC, Superior colliculus. **e**, Spatial distributions of glutamatergic and GABAergic neurons in the thalamus, showing GABAergic neurons in the reticular nucleus (RT) and glutamatergic neurons in the rest of the thalamus. **f**, Spatial distributions of glutamatergic and GABAergic neurons, including the glycinergic neurons, in the midbrain and hindbrain. **g**, Spatial distributions of glutamatergic and GABAergic neurons, including the glycinergic neurons, in the cerebellum. **h**, Spatial distributions of neurons co-expressing *Vglut* (*Slc17a6*, *Slc17a7* or *Slc17a8*) and *Vgat* (*Slc31a1*). AHN, Anterior hypothalamic nucleus; GPi, Globus pallidus, internal segment; SUM, Supramammillary nucleus. **i**, Spatial distributions of neurons expressing *Vglut1* (*Slc17a7*, green) and *Vglut2* (*Slc17a6*, orange). Neurons that co-express *Vglut1* and *Vglut2* are shown in yellow. AOB, Accessory olfactory bulb; AON, Anterior olfactory nucleus; MH, Medial habenula; LD, Lateral dorsal nucleus of thalamus; VPM, Ventral posteromedial nucleus of the thalamus; PG: Pontine gray. **j**–**o**, Spatial distributions of dopaminergic (**j**), serotonergic (**k**), histaminergic (**l**), glycinergic (**m**), noradrenergic (**n**) and cholinergic (**o**) neurons. PVi, Periventricular hypothalamic nucleus, intermediate part; ARH, Arcuate hypothalamic nucleus; VTA, Ventral tegmental area; SNr, Substantia nigra, reticular part; SNc, Substantia nigra, compact part; LDT, Laterodorsal tegmental nucleus; DMH, Dorsomedial nucleus of the hypothalamus; VMH, Ventromedial hypothalamic nucleus; TU, Tuberomammillary nucleus; PMv, Ventral premammillary nucleus; TMv, Tuberomammillary nucleus, ventral part; MV, Medial vestibular nucleus; GRN, Gigantocellular reticular nucleus; RPA, Nucleus raphe pallidus; DCO, Dorsal cochlear nucleus; NTS, Nucleus of the solitary tract; SPVI, Spinal nucleus of the trigeminal, interpolar part; PCG, Pontine central gray; LC, Locus ceruleus; MS, Medial septal nucleus; NDB, Diagonal band nucleus; PBG, Parabigeminal nucleus; PPN, Pedunculopontine nucleus; DMX, Dorsal motor nucleus of the vagus nerve; XII, Hypoglossal nuclei. **p**, Spatial distribution of the inhibitory immature neurons (IMNs) coloured by cluster identities as in Fig. 2f middle panel. Scale bars in **a**–**p**: 1 mm. The underlying contour lines marking brain region boundaries in **a**–**p** were generated using coordinates from the Allen Mouse Brain CCFv3 (ref. 21).

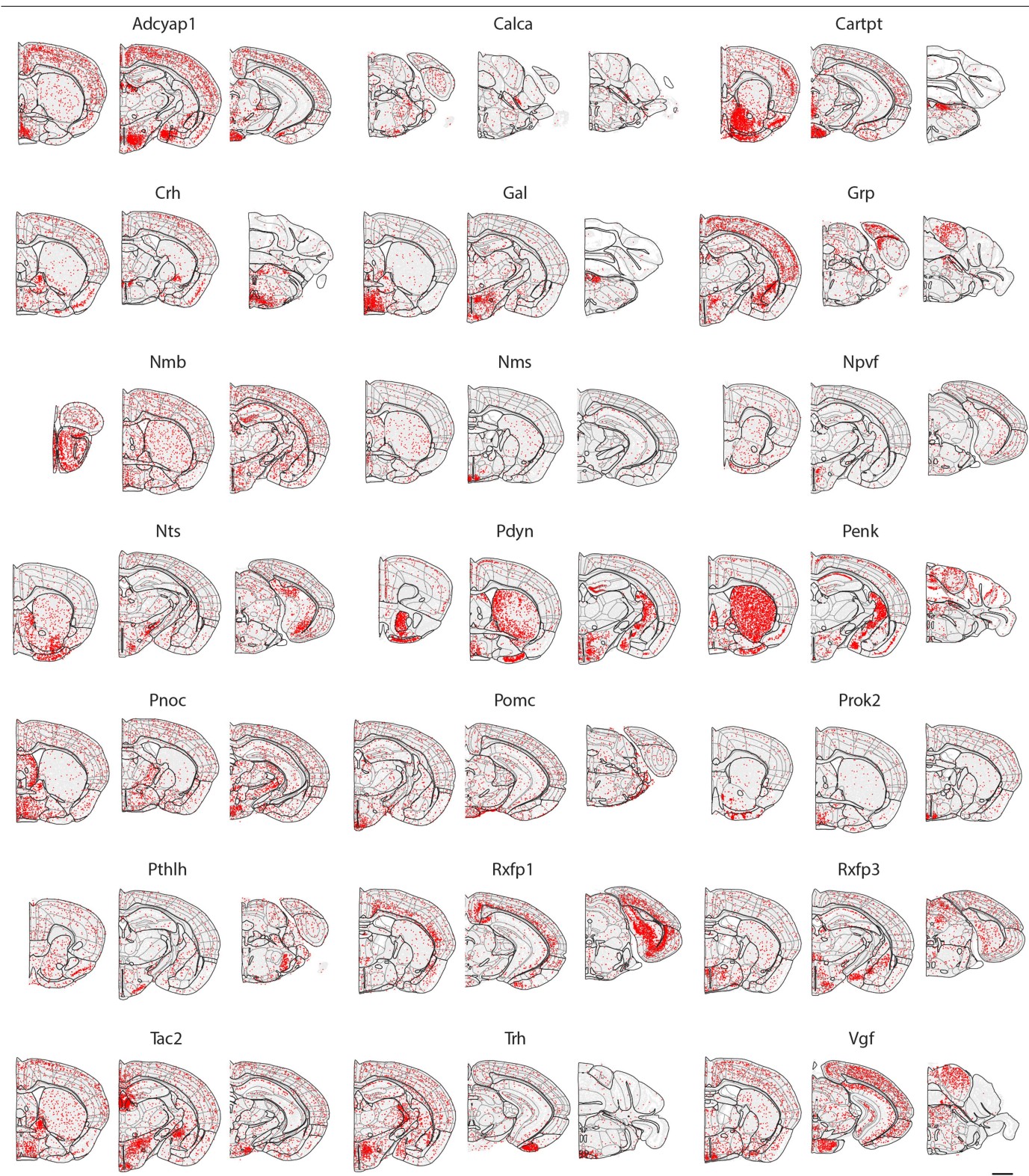

**Extended Data Fig. 5 | Spatial distributions of neuropeptide usage.** Spatial distributions of neurons expressing various neuropeptide genes shown in multiple example coronal slices. Scale bar: 1 mm. The underlying contour lines marking brain region boundaries in the images were generated using coordinates from the Allen Mouse Brain CCFv3 (ref. 21).

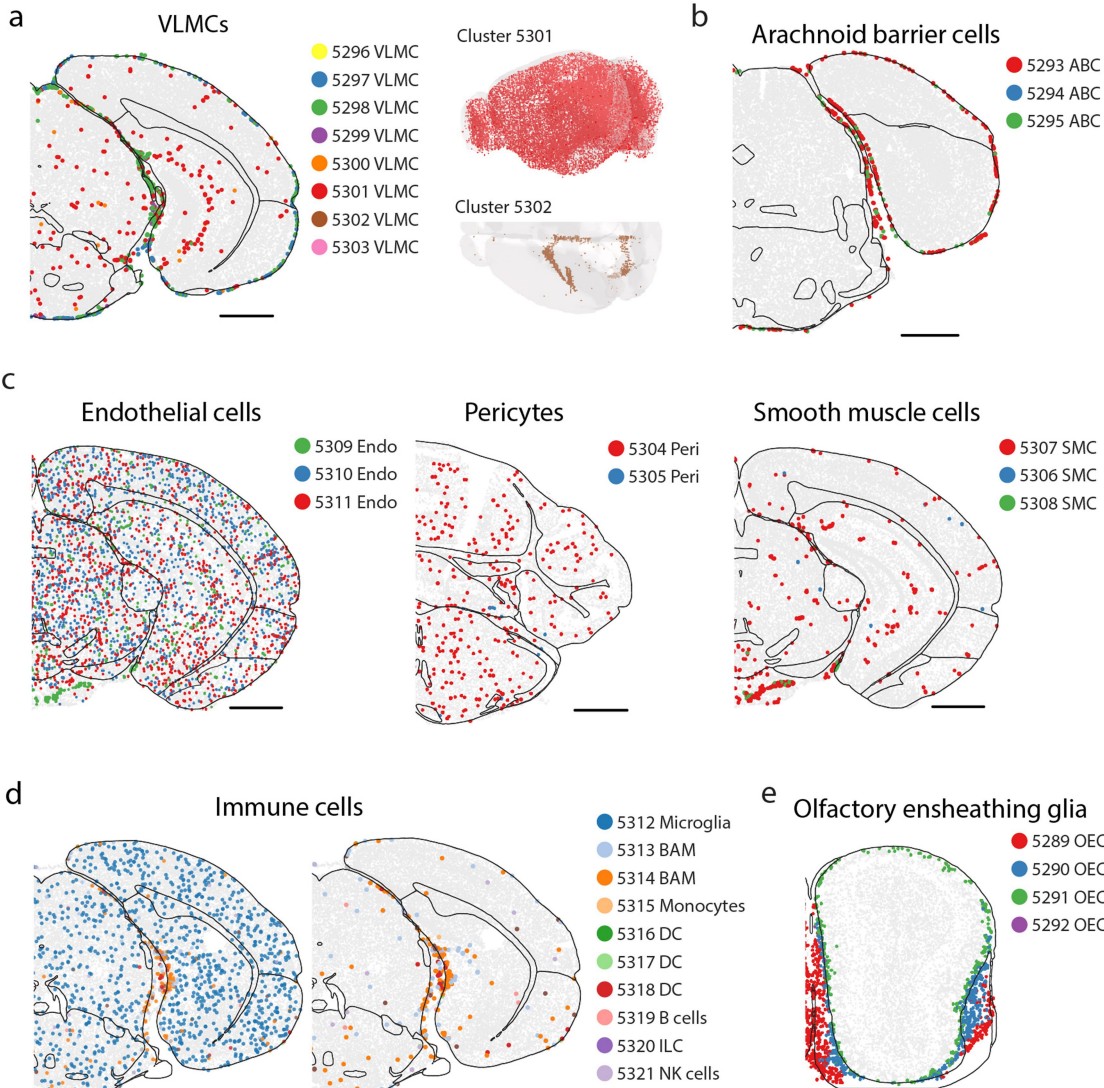

**Extended Data Fig. 6 | Spatial distributions of additional non-neuronal cell types. a**, Left: Spatial distributions of VLMCs shown in an example coronal section. Right: Spatial distributions shown in the 3D CCF space for VLMC cluster 5301 (top), which is enriched in the grey matter, and cluster 5302 (bottom), which is located in the choroid plexus in the lateral and fourth ventricles. **b**, Spatial distributions of arachnoid barrier cells (ABCs) shown in an example coronal section. **c**, Spatial distributions of endothelial cells (left), pericytes (middle) and smooth muscle cells (SMCs, right), each shown in an example coronal section. **d**, Spatial distributions of immune cells shown in an example coronal section including microglia (left) and in the same section but without showing microglia (right). **e**, Spatial distributions of olfactory ensheathing cells (OEC) shown in an example coronal section. Cells are coloured by cluster identities in all panels. Scale bars in **a**–**e**: 1 mm. The underlying contour lines marking brain region boundaries in **a**–**e** and the 3D brain contours in **a** were generated using coordinates from the Allen Mouse Brain CCFv3 (ref. 21).

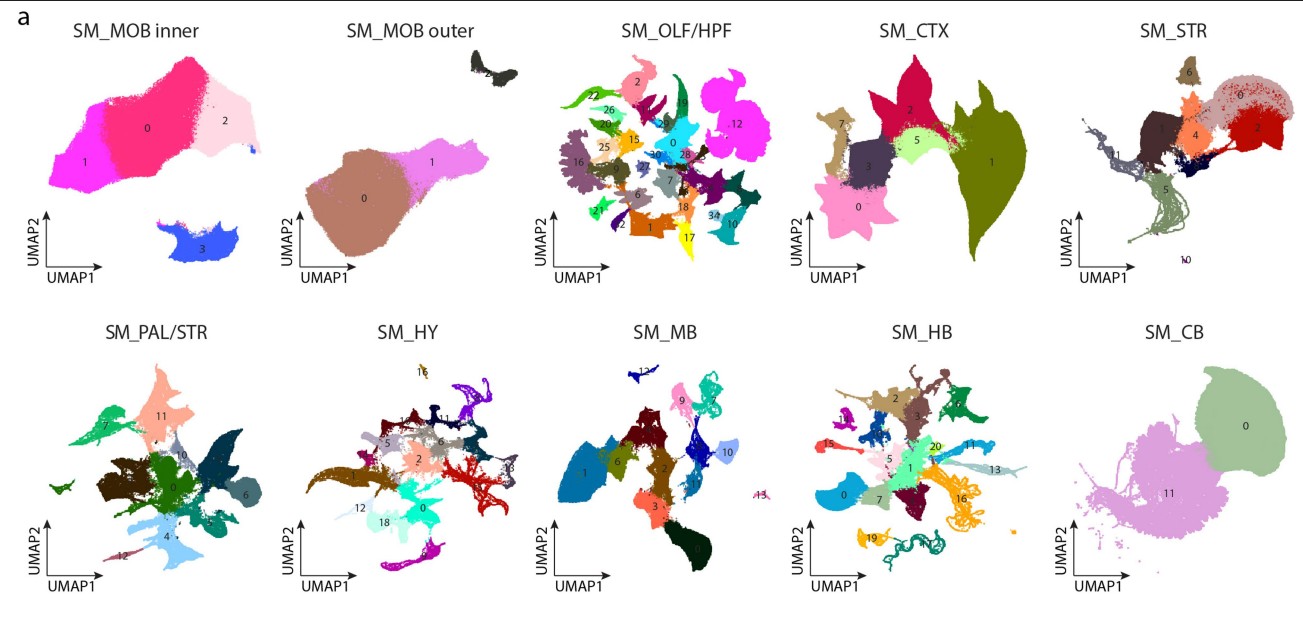

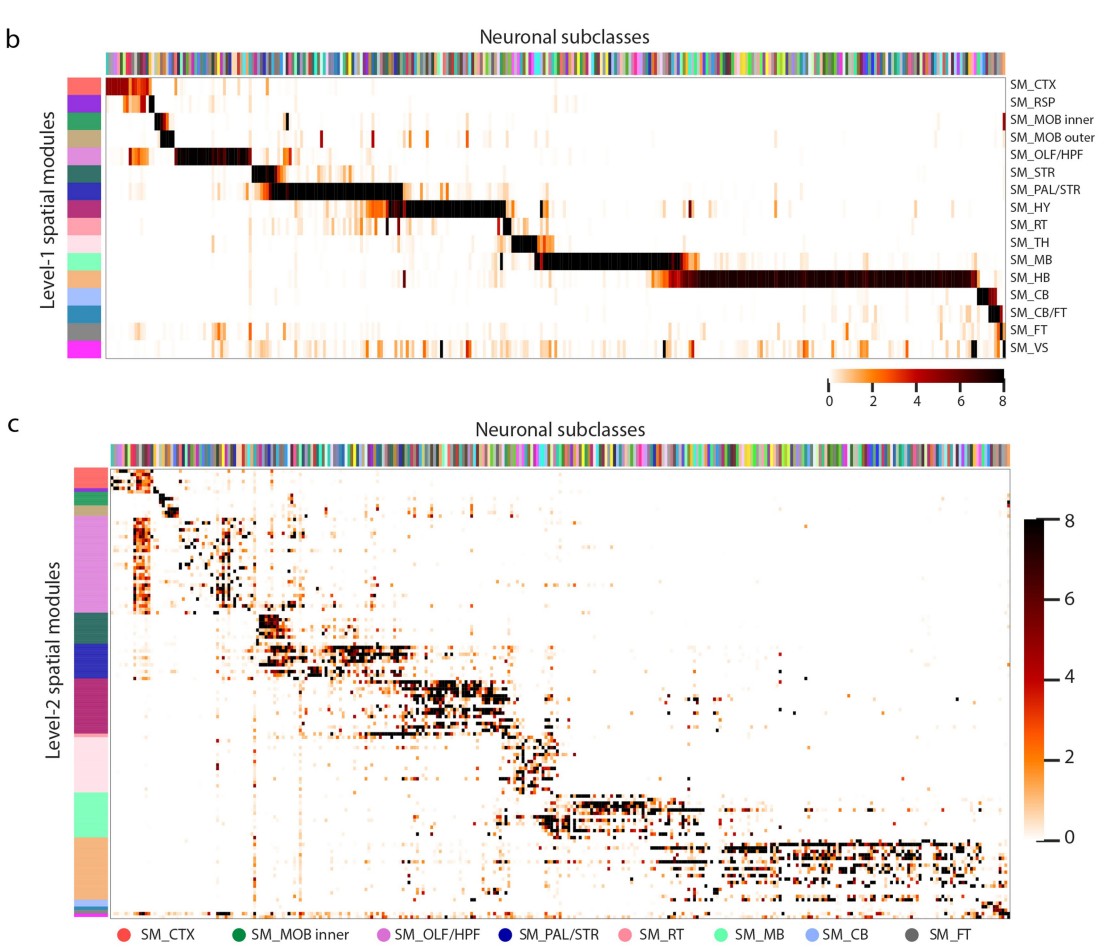

**Extended Data Fig. 7 | Spatial-module delineation. a**, UMAP of cells in the other level-1 spatial module, as in Fig. 4a **bottom**, with cells coloured by their level-2 spatial module identity. **b**,**c**, Heatmaps showing the enrichment scores of all neuronal subclasses in the 16 level-1 spatial modules (**b**) and in the 130 level-2 spatial modules (**c**). The enrichment score is defined as the fold change of the fraction of cells belong to a subclass in each individual spatial module compared to the same fraction across all spatial modules. The coloured bars at the top and on the left indicate the neuronal subclasses and spatial modules, respectively.

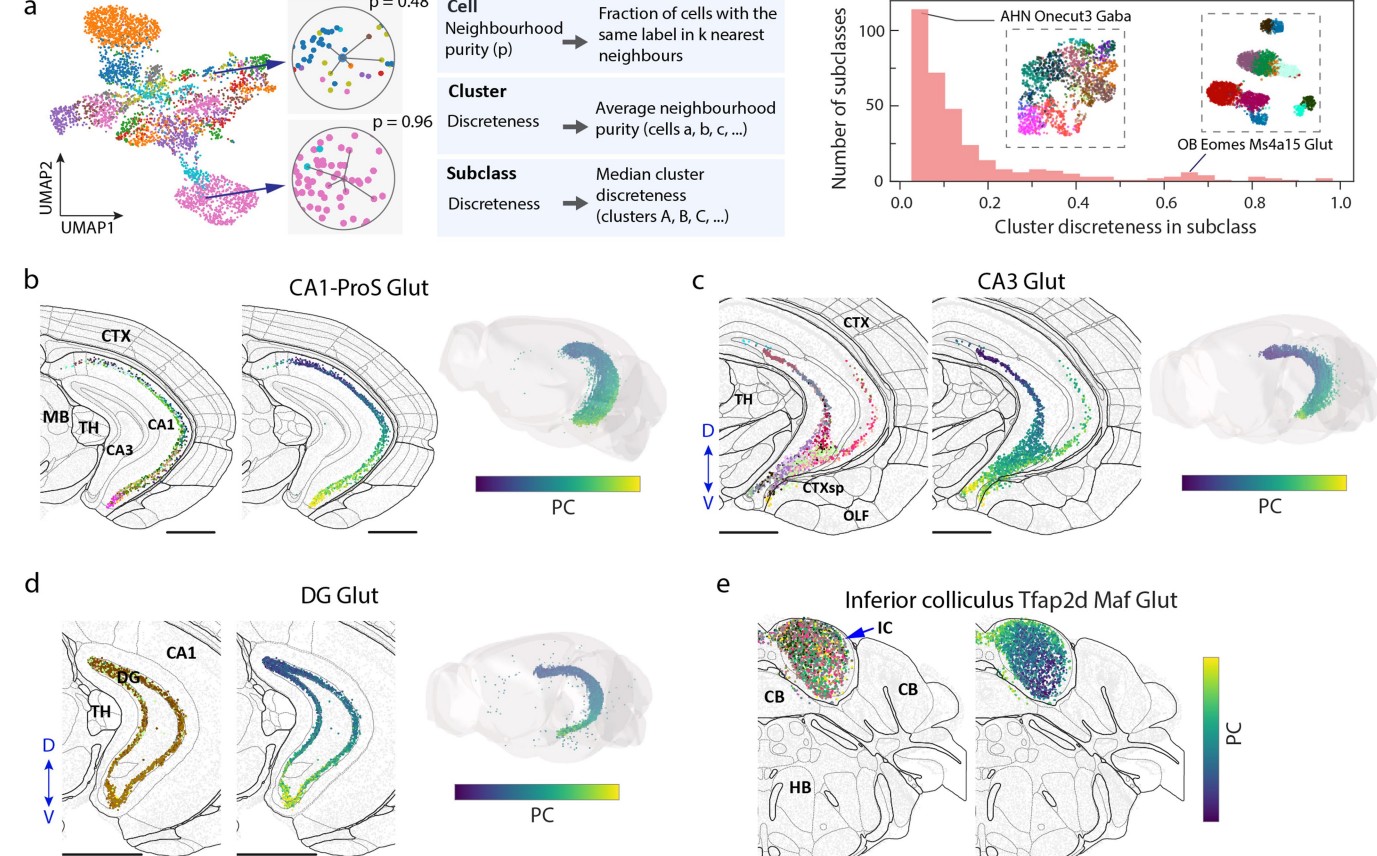

**Extended Data Fig. 8 | Quantification of cluster discreteness of cell subclasses and additional examples of spatial gradients of molecularly defined cell types. a**, Left: To quantify the cluster discreteness in a subclass, a neighbourhood purity quantity for each cell in a cluster is determined as the fraction of the cells in its neighbourhood (in the gene-expression space) that belong to this cluster. The mean neighbourhood purity quantity across all cells in a cluster is defined as the discreteness of the cluster, which gives a measure of how well separated this cluster is from the other clusters in the gene-expression space. The median discreteness of clusters is then determined for each subclass. Right: Distribution of the median cluster discreteness of individual subclasses across all subclasses. The UMAPs of an example subclass with high cluster discreteness (OB Eomes Ms4a15 Glut) and an example subclass

with low cluster discreteness (AHN Onecut3 Gaba) are shown. **b–d**, Spatial gradients of CA1-Pros Glut neurons (**b**), CA3 Glut neurons (**c**) and DG Glut neurons (**d**) in the hippocampal formation. From left to right: Spatial map of cells coloured by cluster identities in a coronal section; Spatial map of cells coloured by the first principal component (PC1) in the same section; Spatial distribution of cells colored by PC1 shown in the 3D CCF space. **e**, Spatial gradient of the Tfap2d Maf Glut neurons in the inferior colliculus (IC) of the midbrain. Cells are shown in one coronal section and are coloured by cluster identities (left) and PC1 (right). Scale bars in **b**–**e**: 1 mm. The underlying contour lines marking brain region boundaries in **b**–**e** and the 3D brain contours in **b**–**d** were generated using coordinates from the Allen Mouse Brain CCFv3 (ref. 21).

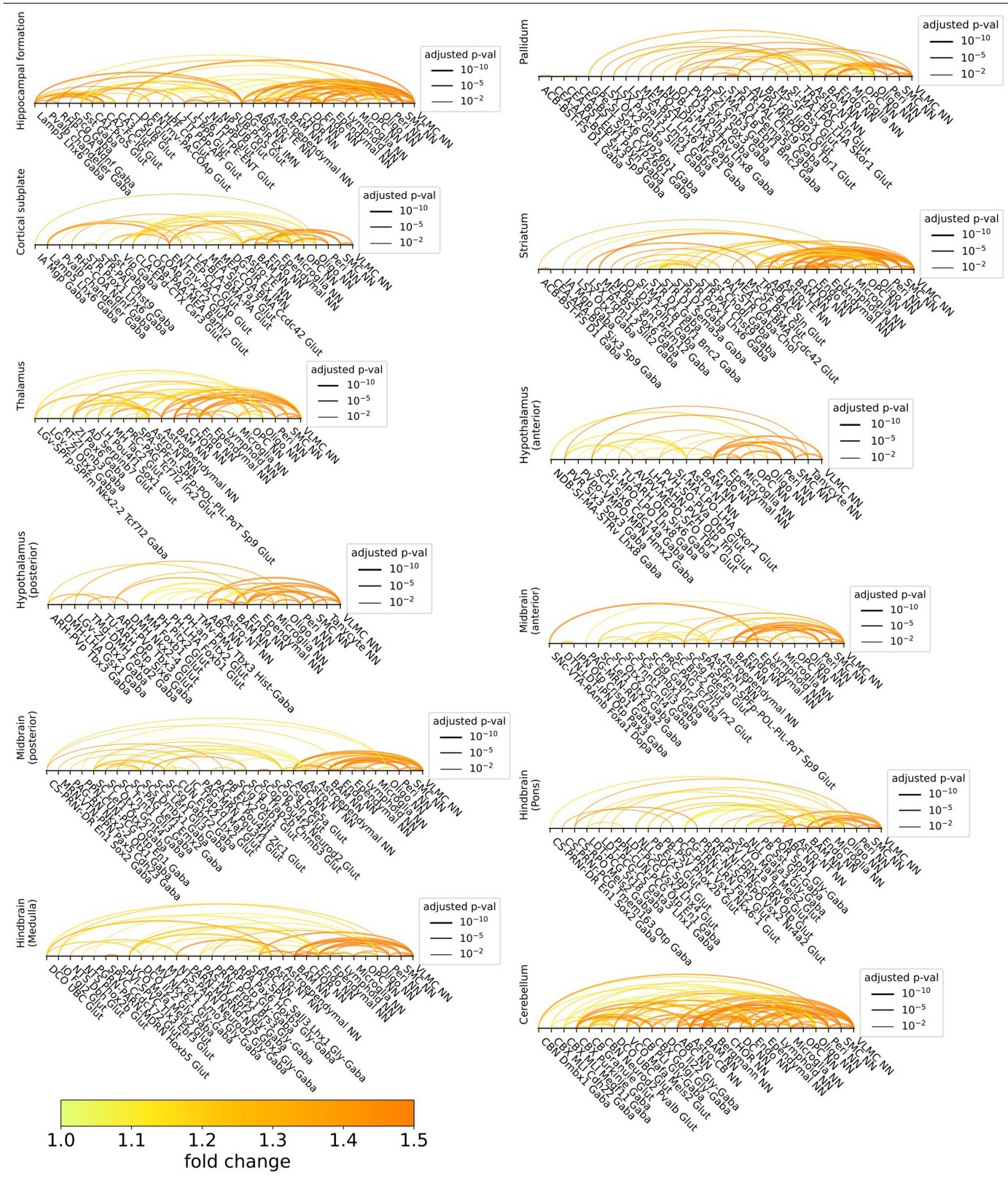

**Extended Data Fig. 9 | Predicted cell-cell interactions or communications in individual brain regions.** Same as in Fig. 6c, but for the hippocampal formation, cortical subplate, striatum, pallidum, thalamus, hypothalamus (anterior and posterior parts), midbrain (anterior and posterior parts), hindbrain (pons and medulla sub-regions), and cerebellum.

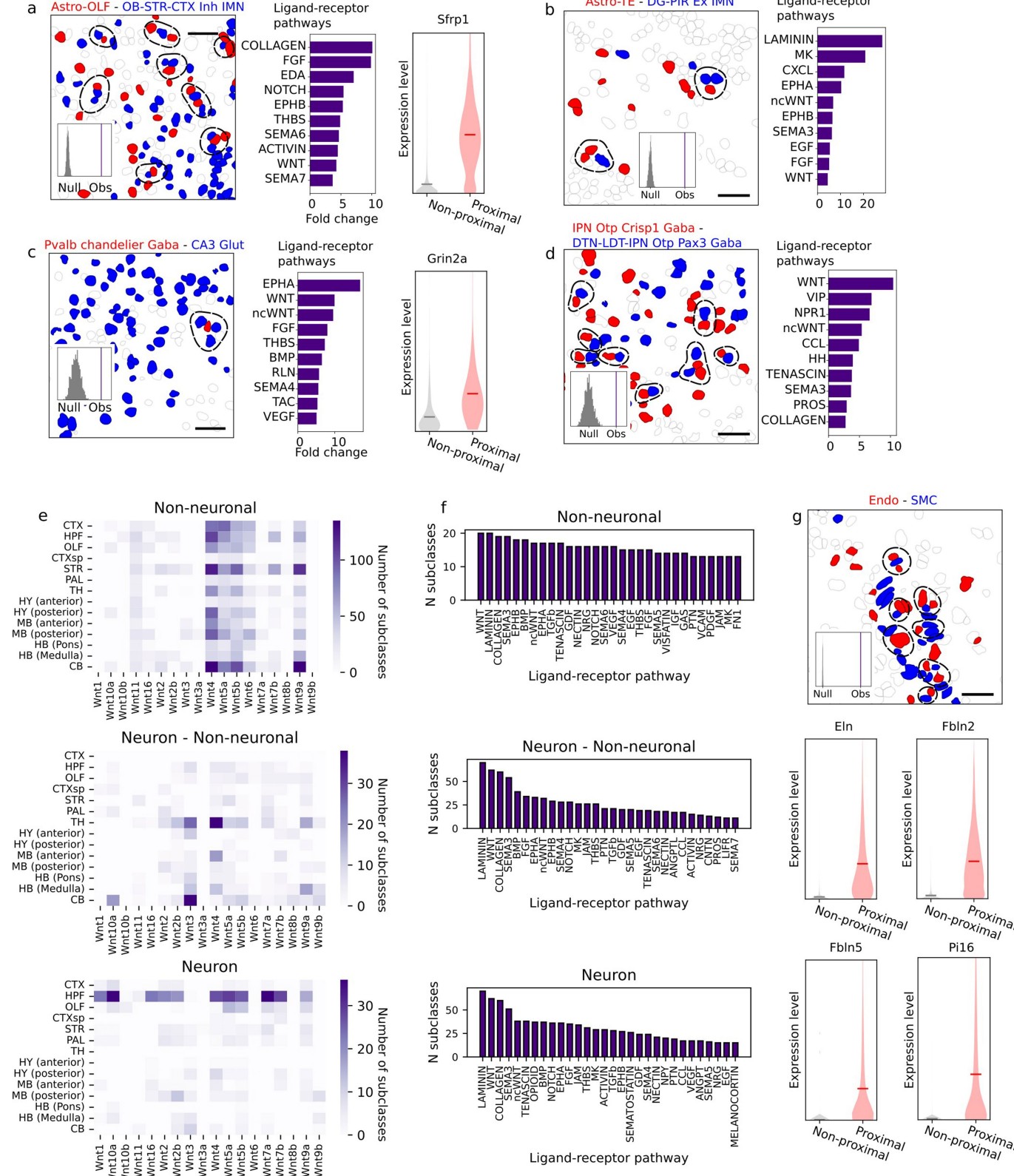

**Extended Data Fig. 10** | See next page for caption.

**Extended Data Fig. 10 | Additional examples and characterizations of predicted cell-cell interactions or communications. a**, Interactions between olfactory astrocytes (Astro-OLF) and inhibitory immature neurons (MOB-STR-CTX inh IMN). Left: Example image of cells in a small area, with cells belonging to the indicated cell types shown in red and blue and all other cells shown in grey, as described in Fig. 6d. Middle: Top 10 upregulated ligand-receptor pathways, as described in Fig. 6d. Right: Expression distributions of the indicated gene in Astro-OLF proximal (red) or non-proximal (grey) to MOB-STR-CTX inh IMN, as described in Fig. 6d. **b**–**d**, Same as **a**, but for interactions between astrocytes (Astro-TE) and excitatory immature neurons (DG-PIR Ex IMN) (**b**), between Pvalb chandelier Gaba neurons and CA3 Glut neurons (**c**), and between IPN Otp Crisp1 Gaba neurons and DTN-LDT-IPN Otp Pax3 Gaba neurons (**d**). In (**b**) and (**d**), violin plots of example genes upregulated in proximal cell pairs as compared to non-proximal cell pairs are not shown. **e**, Total numbers of unique cell types (subclasses) observed in the interacting cell-type pairs that showed upregulation of the ligand-receptor pairs involving the indicated Wnt ligands in each of the major brain regions. Top: For interactions among non-neuronal cells; Middle: For interactions between neurons and non-neuronal cells; Bottom: For interactions among neurons. **f**, The total number of unique cell-types (subclasses) involved in the predicted interacting cell-type pairs that showed upregulation of ligand-receptor pairs in the indicated pathway across the whole brain. For each category of cell-cell interactions (interactions among non-neuronal cells (top), interactions between neurons and non-neuronal cells (middle), and interactions among neurons (Bottom)), the top 30 ligand-receptor pathways with the highest number of cell types involved are shown. **g**, Interactions between endothelial cells and SMC cells. Top: Example image of cells in a small area, as described in Fig. 6d. Bottom: Expression distributions of the indicated genes in endothelial cells when they are proximal or non-proximal to SMC. Scale bars in **a**,**b**,**e**: 30 μm.

# Reporting Summary

## Statistics

For all statistical analyses, confirm that the following items are present in the figure legend, table legend, main text, or Methods section.

| n/a | Confirmed | |
|---|---|---|
| ☐ | ☒ | The exact sample size (*n*) for each experimental group/condition, given as a discrete number and unit of measurement |
| ☐ | ☒ | A statement on whether measurements were taken from distinct samples or whether the same sample was measured repeatedly |
| ☐ | ☒ | The statistical test(s) used AND whether they are one- or two-sided <br> *Only common tests should be described solely by name; describe more complex techniques in the Methods section.* |
| ☒ | ☐ | A description of all covariates tested |
| ☐ | ☒ | A description of any assumptions or corrections, such as tests of normality and adjustment for multiple comparisons |
| ☐ | ☒ | A full description of the statistical parameters including central tendency (e.g. means) or other basic estimates (e.g. regression coefficient) AND variation (e.g. standard deviation) or associated estimates of uncertainty (e.g. confidence intervals) |
| ☐ | ☒ | For null hypothesis testing, the test statistic (e.g. *F*, *t*, *r*) with confidence intervals, effect sizes, degrees of freedom and *P* value noted <br> *Give P values as exact values whenever suitable.* |
| ☒ | ☐ | For Bayesian analysis, information on the choice of priors and Markov chain Monte Carlo settings |
| ☒ | ☐ | For hierarchical and complex designs, identification of the appropriate level for tests and full reporting of outcomes |
| ☐ | ☒ | Estimates of effect sizes (e.g. Cohen's *d*, Pearson's *r*), indicating how they were calculated |

*Our web collection on statistics for biologists contains articles on many of the points above.*

## Software and code

Policy information about availability of computer code

| | |
|---|---|
| Data collection | MERFISH imaging data was collected using custom Python code to control the microscope. This code is available at https://github.com/ZhuangLab and on Zenodo. |
| Data analysis | The MERFISH data was analyzed using custom Python code. Code for MERFISH image analysis is available at https://github.com/ZhuangLab/MERlin and on Zenodo. Additional code for data analysis is available at https://github.com/ZhuangLab/whole_mouse_brain_MERFISH_atlas_scripts_2023 and on Zenodo. <br> Other packages used in data analyses include: Cellpose (version 2.0); Scanpy (version 1.9.1); Scrublet (version 0.2); ALLCools (version 0.2.19); metis (version 0.2a5); scikit-learn (version 1.1.1); Elastix (version 5.1.0); Brainrender (version 2.0.0.0). |

For manuscripts utilizing custom algorithms or software that are central to the research but not yet described in published literature, software must be made available to editors and reviewers. We strongly encourage code deposition in a community repository (e.g. GitHub). See the Nature Portfolio guidelines for submitting code & software for further information.

# Data

Policy information about availability of data

All manuscripts must include a data availability statement. This statement should provide the following information, where applicable:
- Accession codes, unique identifiers, or web links for publicly available datasets
- A description of any restrictions on data availability
- For clinical datasets or third party data, please ensure that the statement adheres to our policy

Data availability statement is included in the manuscript, which states:

Raw and processed MERFISH data, as well as the MERFISH codebook and probes used in this work, can be accessed via the Brain Image Library (BIL): https://doi.org/10.35077/act-bag . Processed MERFISH data are also accessible and explorable in an interactive manner through two platforms: 1. Allen Brain Cell (ABC) Atlas (https://knowledge.brain-map.org/data/5C0201JSVE04WY6DMVC/explore; https://allen-brain-cell-atlas.s3.us-west-2.amazonaws.com/index.html); 2. CELLxGENE database (https://cellxgene.cziscience.com/collections/0cca8620-8dee-45d0-aef5-23f032a5cf09).

The scRNA-seq datasets (FASTQ files) obtained by Allen Institute are available at NeMO under identifier https://assets.nemoarchive.org/dat-qg7n1b0. The processed scRNA-seq data along with the transcriptomic cell type taxonomy is visualized at ABC Atlas – mouse whole brain cell type atlas, https://portal.brain-map.org/atlases-and-data/bkp/abc-atlas. Instruction for access of the processed scRNA-seq data is available at https://github.com/AllenInstitute/abc_atlas_access/blob/main/descriptions/WMB-10X.md.

CellChat database is available at (http://www.cellchat.org/).

# Human research participants

Policy information about studies involving human research participants and Sex and Gender in Research.

| Reporting on sex and gender | Not applicable |
|---|---|
| Population characteristics | Not applicable |
| Recruitment | Not applicable |
| Ethics oversight | Not applicable |

Note that full information on the approval of the study protocol must also be provided in the manuscript.

# Field-specific reporting

Please select the one below that is the best fit for your research. If you are not sure, read the appropriate sections before making your selection.

☒ Life sciences        ☐ Behavioural & social sciences        ☐ Ecological, evolutionary & environmental sciences

For a reference copy of the document with all sections, see nature.com/documents/nr-reporting-summary-flat.pdf

# Life sciences study design

All studies must disclose on these points even when the disclosure is negative.

| Sample size | Four replicate mice, one female and three males, were imaged under each condition. From the four replicate mice imaged for the identification and spatial mapping of cell types, a total of approximately 10 million cells were imaged and segmented, which generated a sufficient number of single-cell profiles and gave sufficient statistics for the effect sizes of interest. No statistical methods were used to predetermine sample size and sample size were determined empirically. |
|---|---|
| Data exclusions | We did not exclude any data from consideration. All images were included in the primary analysis. |
| Replication | Reported results were replicated from four animals under each condition. |
| Randomization | Four animals, one female and three males, were randomly chosen for the identification and spatial mapping of cell types. For each mouse, the imaging experiments were definitive, and no randomization was necessary for this study, hence the experiments were not randomized. Animals were not allocated into experimental groups. |
| Blinding | The investigators were not blinded during experiments and outcome assessment. Blinding during data collection was not needed because all images were taken under same condition. Blinding during analysis was not necessary because the results were quantitative and did not require subjective judgment. Blinding is not typically used in the field. |

# Reporting for specific materials, systems and methods

We require information from authors about some types of materials, experimental systems and methods used in many studies. Here, indicate whether each material, system or method listed is relevant to your study. If you are not sure if a list item applies to your research, read the appropriate section before selecting a response.

## Materials & experimental systems

| n/a | Involved in the study |
|-----|----------------------|
| ☒ | ☐ Antibodies |
| ☒ | ☐ Eukaryotic cell lines |
| ☒ | ☐ Palaeontology and archaeology |
| ☐ | ☒ Animals and other organisms |
| ☒ | ☐ Clinical data |
| ☒ | ☐ Dual use research of concern |

## Methods

| n/a | Involved in the study |
|-----|----------------------|
| ☒ | ☐ ChIP-seq |
| ☒ | ☐ Flow cytometry |
| ☒ | ☐ MRI-based neuroimaging |

## Animals and other research organisms

Policy information about studies involving animals; ARRIVE guidelines recommended for reporting animal research, and Sex and Gender in Research

| | |
|---|---|
| Laboratory animals | Adult C57BL/6NCrl (Strain code: 027, Charles River Laboratories) male and female mice aged 56-62 days were used in this study. Animals were purchased from the Charles River Laboratories at an age one week younger (49-55 days) than the target age for sacrifice and housed at Harvard University Animal Facility for 1 week to acclimate before sacrifice. Mice were maintained on a 12 hour:12 hour light/dark cycle (2pm-2am dark period) with at a temperature of 22 ± 1°C, a humidity of 30–70%, with ad libitum access to food and water. All the animals used in this study were sacrificed between 2-6pm of the day. |
| Wild animals | The study did not involve wild animals. |
| Reporting on sex | Four animals were used in this study, including one female mice and three male mice. No sex-specific results are reported. |
| Field-collected samples | The study did not involve samples collected from the field. |
| Ethics oversight | Harvard University Institutional Animal Care and Use Committee |

Note that full information on the approval of the study protocol must also be provided in the manuscript.

