## [Peer Review File · Nature]

Manuscript Title: Molecularly defined and spatially resolved cell atlas of whole mouse brain

Reviewer Comments & Author Rebuttals

Reviewer Reports on the Initial Version:

Referees' comments:

Referee #1 (Remarks to the Author):

This landmark study by the Zhuang lab maps thousands of transcriptionally distinct types and 300 major cell types across the entire mouse brain. They used the now well-established MERFISH technique, and this study proves that it is an incredibly powerful workhorse for large scale spatial transcriptomics in brain tissue. The MERFISH results were registered to the Allen common coordinate framework, revealing the anatomical organization of the mouse brain neuronal and non-neuronal cell types. Importantly, they used this data to define molecularly distinct subregions of the mouse brain and to identify cell type gradients as well. The anatomical elements of this study are simply invaluable to the neuroscience community.

As such, my main critique is the relative lack of accessibility to all of these important details. The manuscript necessarily covers a few general points at a high level, but neither the figures or the text delve into the details in a way that would be useful to a neuroscience researcher. The Supplementary tables are comprehensive and the data is available on the Brain Image Library but these formats are not conducive to exploring this spatial information without intensive time and effort. To have the appropriate impact, it is important for this atlasing effort that the authors provide a well annotated tool for interacting in detail with this data, for example to generate the images associated with the atlas overlays of cell types or spatially defined brain regions.

Referee #2 (Remarks to the Author):

This is a data resource paper that generated a molecularly defined and spatially resolved cell atlas of the whole mouse brain. This resource uses four adult mice and samples across the entire brain using both coronal and sagittal sections. The resource is a combination of in situ sequencing using MERFISH with ~1K genes combined with single cell expression profiling with genome wide coverage. The project is part of an initiative funded by the BRAIN Initiative Cell Census Network (BICCN). The funding mechanism for this project (RFA-MH-17-225) was part of a large and expensive effort to generate molecular profiles of cell types across the brain as resources for the neuroscience community. The data acquired here is of high quality, important and represents an enormous time and financial investment. I have only minor issues/questions about the data acquisition, quality of the data, logistics of acquisition. However, the main issue with the paper is related to data accessibility. As a community resource, I do not feel that the data is highly accessible and useable to the broader neuroscience community. Given that the data acquisition was undertaken with the goal of being a resource, the paper and its presentations falls short in that what is presented in the paper barely scratches the surface of what the data can be used for. The data can be mined in many ways and could be extremely useful. However, without an easy to use visualization application, web browser designed for novice data scientists, I find that the data set, as a resource, is much less compelling. While the data is accessible here:

<https://doi.brainimagelibrary.org/doi/10.35077/act-bag> it is not provided or presented in a way that would be easily navigable or useable for most users that are not already familiar with the data types. If someone wants to interrogate the data for their own questions or purposes, I do not feel that this would be easy for most novice users to do. Given that the novelty and importance of this paper is mainly the data itself as a resource (size, scope and breadth across brain) rather than any biological finding per se, I feel that this is not appropriate – as a resource the data needs to be more accessible.

Minor comments:

1. Additional information on the animals is necessary

- a. Substrain of the C57Bl6 mice should be reported (e.g. N, J?)
- b. Were the animals ordered directly from a vendor or were they bred in house?
- c. It would be better to put the age in the "Animals" subsection – I found it somewhere else in the methods
- d. What time of day were the four animals killed at and was this consistent across all animals?

2. More information about the panel selection would be helpful. There are some discrepancies that are not clear.

- a. Why the panels for the first animal differed from the others is not clear. What was the rationale for this change? Why did it happen?
- b. The method for gene inclusion in the panel based on DE analysis is clear in the methods – but, how were genes that were chosen/prioritized and trimmed down/selected to the max available space (e.g. ~1K)?
- c. What was the rationale for the addition of Sst, Vip, Avp and Pmch?

3. In Figure 1, the color classes are labeled for "a", but there are no corresponding color legends (on the figure or in the legend) for the subclasses in c-d, which makes the figure not as useful as it could be since one can't assess what those sub-classes actually are.

Author Rebuttals to Initial Comments:

Point-by-point response to reviewer comments

Referee #1 (Remarks to the Author):

This landmark study by the Zhuang lab maps thousands of transcriptionally distinct types and 300 major cell types across the entire mouse brain. They used the now well-established MERFISH technique, and this study proves that it is an incredibly powerful workhorse for large scale spatial transcriptomics in brain tissue. The MERFISH results were registered to the Allen common coordinate framework, revealing the anatomical organization of the mouse brain neuronal and non-neuronal cell types. Importantly, they used this data to define molecularly distinct subregions of the mouse brain and to identify cell type gradients as well. The anatomical elements of this study are simply invaluable to the neuroscience community.

As such, my main critique is the relative lack of accessibility to all of these important details. The manuscript necessarily covers a few general points at a high level, but neither the figures or the text delve into the details in a way that would be useful to a neuroscience researcher. The Supplementary tables are comprehensive and the data is available on the Brain Image Library but these formats are not conducive to exploring this spatial information without intensive time and effort. To have the appropriate impact, it is important for this atlas effort that the authors provide a well annotated tool for interacting in detail with this data, for example to generate the images associated with the atlas overlays of cell types or spatially defined brain regions.

Response: We thank the reviewer for his/her enthusiasm about our work and for the comment on the accessibility of the data. We also recognize the importance of making our data available to the scientific community in an accessible and interactive way and indeed planned to do that before the paper is published. We are happy to report that we have now made our data accessible and explorable in an interactive manner through two platforms, the Allen Brain Cell Atlas database and the CellxGene database.

First, we have made our data accessible and explorable through the Allen Brain Cell Atlas database (URL for interactive exploration of our data: <https://knowledge.brain-map.org/data/5C0201JSVE04WY6DMVC/explore>; URL for downloading our data: <https://allen-brain-cell-atlas.s3.us-west-2.amazonaws.com/index.html>). In this platform, users will be able to visualize and interact with the whole mouse brain MERFISH datasets that we generated. In these datasets, we provided the cell type identity, the spatial coordinates in the Allen Mouse Brain Common Coordinate Framework (CCF), and the expression levels of the genes measured by MERFISH for each individual imaged cell.

Below are some of important features of what users can do with our MERFISH data on this platform:

- Plot spatial maps of the whole brain tissue slices and color the cells by cell types at different levels.
- Search for gene of interest and color cells in the spatial maps based on the expression levels of the gene of interest.

- Select the cells of specified cell types and/or cells based on the neurotransmitter usage, and highlight these cells on the spatial maps.
- Select brain regions of interest based on Allen Mouse CCF and highlight the cells that fall within the region.
- Select cells by the CCF regions and color cells by cell types or gene expression to facilitate visual inspection of cell-type compositions and gene-expression patterns in individual brain regions.

These are functions included in the initial release of the Allen Brain Cell Atlas platform. We also anticipate that additional useful features of data exploration will be included in this platform in the future based on user feedback.

Second, we also uploaded our data, including both the MERFISH measured gene expression profiles of individual cells and the imputed transcriptome-wide expression profiles of individual cells (determined by integration of our MERFISH data with scRNA-seq data obtained by the Allen Institute), to the CELLxGENE database (URL for our data: <https://cellxgene.cziscience.com/collections/0cca8620-8dee-45d0-aef5-23f032a5cf09>).

Below are some of important features of what the users can do with our MERFISH data on the CELLxGENE platform:

- Plot UMAP and spatial maps of our whole brain dataset and color cells by their cell type classifications.
- Search for gene of interest and color or filter cells based on the expression levels of the gene of interest.
- Select the cells of interest based on their cell type classifications and highlight these cells on the UMAP or spatial maps.
- Select brain region of interest based on Allen Mouse CCF and highlight cells that fall within the region.
- Select cells by the CCF regions and color cells by cell types or gene expression to facilitate visual inspection of cell-type compositions and gene-expression patterns in individual brain regions.
- Filter cells by cell-type classification confidence score.

By making our data accessible and explorable through these publicly available and interactive platforms, we think our data can be used by anyone interested in exploring our data and thus provide a valuable resource for the neuroscience community, and more broadly, the whole scientific community.

Referee #2 (Remarks to the Author):

This is a data resource paper that generated a molecularly defined and spatially resolved cell atlas of the whole mouse brain. This resource uses four adult mice and samples across the entire brain using both coronal and sagittal sections. The resource is a combination of in situ sequencing using MERFISH with ~1K genes combined with single cell expression profiling with genome wide coverage. The project is part of an initiative funded by the BRAIN Initiative Cell Census Network (BICCN). The funding mechanism for this project (RFA-MH-17-225) was part of a large and expensive effort to generate molecular profiles of cell types across the brain as resources for the neuroscience community. The data acquired here is of

high quality, important and represents an enormous time and financial investment. I have only minor issues/questions about the data acquisition, quality of the data, logistics of acquisition. However, the main issue with the paper is related to data accessibility. As a community resource, I do not feel that the data is highly accessible and useable to the broader neuroscience community. Given that the data acquisition was undertaken with the goal of being a resource, the paper and its presentations falls short in that what is presented in the paper barely scratches the surface of what the data can be used for. The data can be mined in many ways and could be extremely useful. However, without an easy to use visualization application, web browser designed for novice data scientists, I find that the data set, as a resource, is much less compelling. While the data is accessible

here: <https://doi.brainimagedlibrary.org/doi/10.35077/act-bag> it is not provided or presented in a way that would be easily navigable or useable for most users that are not already familiar with the data types. If someone wants to interrogate the data for their own questions or purposes, I do not feel that this would be easy for most novice users to do. Given that the novelty and importance of this paper is mainly the data itself as a resource (size, scope and breadth across brain) rather than any biological finding per se, I feel that this is not appropriate – as a resource the data needs to be more accessible.

Response: We thank the reviewer for his/her enthusiasm about our work and for the comment on the accessibility of the data. We also recognize the importance of making our data available to the scientific community in an accessible and interactive way, and indeed planned to do that before the paper is published. We are happy to report that we have now made our data accessible and navigable in an interactive manner through two platforms, the Allen Brain Cell Atlas database and the CellxGene database. Please see details in our response to Review 1's similar comment above. By making our data accessible and navigable through these publicly available and interactive platforms, we think our data can be used by anyone interested in exploring our data and thus provide a valuable resource for the neuroscience community, and more broadly, the whole scientific community.

Minor comments:

1. Additional information on the animals is necessary

a. Substrain of the C57Bl6 mice should be reported (e.g. N, J?)

Response: The substrain of mice used in this study is C57BL/6NCrl. We have made it clear now in the "Animals" section in Methods.

b. Were the animals ordered directly from a vendor or were they bred in house?

Response: The animals were ordered directly from a vendor (Charles River Laboratories). We have added this information to the "Animals" section in Methods.

c. It would be better to put the age in the "Animals" subsection – I found it somewhere else in the methods

Response: The age information is included in the "Animals" section in Methods.

d. What time of day were the four animals killed at and was this consistent across all animals?

Response: All four animals studied in this work were sacrificed between 2-6pm in the afternoon and we have now added this information in the “Animals” section in the Methods.

2. More information about the panel selection would be helpful. There are some discrepancies that are not clear.

Response: As suggested, we have included more detailed information about the gene-panel selection in “Gene selection for MERFISH” section in Methods to provide further clarity on this topic. Please see details below.

a. Why the panels for the first animal differed from the others is not clear. What was the rationale for this change? Why did it happen?

Response: Indeed, we have imaged the first animal (referred to as Animal #2 in the manuscript) using one gene panel (1124 genes) and the other three animals (referred to as Animal #1, #3, and #4) using a second gene panel (1147 genes). These two gene panels are very similar to each other. In the second gene panel, we added 25 manually picked genes, including additional cell-type markers for non-neuronal cells, additional neurotransmitter-related genes, and neuropeptide genes, and removed two genes (*Nrgn* and *Mag*) from the first gene panel. The two gene panels have 98% of the genes (1122 genes) in common, and only the 1122 common genes from both panels were used to integrate the MERFISH data with scRNA-seq data for cell-type classification. Historically, animal #2 were imaged first and we made the changes in the gene panel after collecting the majority of the data of this animal. Since the 1122 common genes were sufficient for cell-type classification (it allowed us to integrate MERFISH and scRNA-seq data and transfer cell-type labels of all 338 subclasses and ~99% of the 5322 cell clusters from the scRNA-seq data to MERFISH data with high confidence), we decided to keep all data from the first imaged animal and used the 1122 common genes that are present in the data from all animals for the cell-type classification purpose.

The 25 genes that we have added to the second gene panel were mostly good marker genes for specific cell types, and have been extensively studied. Therefore, although not being used for cell-type classification, these 25 genes can provide useful information for readers. For example, the spatial distributions of these genes could be of interest to neurobiologists who study these genes or the specific cell types that these genes mark.

We have now provided these details in “Gene selection for MERFISH” section in Methods.

b. The method for gene inclusion in the panel based on DE analysis is clear in the methods – but, how were genes that were chosen/prioritized and trimmed down/selected to the max available space (e.g. ~1K)?

Response: As suggested, we added more information, as described below, in the “Gene selection for MERFISH” section in Methods to provide further clarity on this topic. The MERFISH gene panel was

constructed through selection of differentially expressed (DE) genes that met the length and abundance requirements as described in this section. The selection was based on the significance of these genes in neuroscience studies, and their effectiveness in distinguishing different cell clusters. Our selection process began with 123 subclass markers that have been defined based on scRNA-seq clustering results. We then continued to add DE genes that also fell into the categories of transcription factors, neural peptides, G-protein-coupled receptors (GPCRs), interleukins, and secreted proteins (229 genes in total). Following this, we employed a greedy search algorithm to iteratively add genes that have the most potent discriminative power in distinguishing pairs of cell clusters that were not adequately separated by the already chosen genes. This greedy search was concluded once there were at least three DE genes included for each pair of clusters in each direction, which in total added up to ~1100 genes. Finally, we added some manually picked genes of interest, such as a few circadian clock genes, previously known non-neuronal marker genes, neurotransmitter-related genes, neuropeptide genes, etc., to form the final gene panels.

c. What was the rationale for the addition of Sst, Vip, Avp and Pmch?

Response: Although these four genes did not meet the length and abundance requirements that we set for genes to be imaged by the combinatorial MERFISH approach, these genes were included because they were classified as subclass markers based on the scRNA-seq data. Therefore, we added these genes and imaged them in the sequential imaging rounds after the combinatorial MERFISH run. We have added this clarification in the “Gene selection for MERFISH” section in Methods.

3. In Figure 1, the color classes are labeled for “a”, but there are no corresponding color legends (on the figure or in the legend) for the subclasses in c-d, which makes the figure not as useful as it could be since one can’t assess what those sub-classes actually are.

Response: The >300 cell subclasses made it difficult to include the color legends in panels (c-d). In our view, these are illustrative figures that do not serve the purpose of showing which cell types are where in the brain, but to convey the point that we identified an extremely high diversity of cells and determined their spatial organizations across the whole brain. As mentioned above in our response to the data-accessibility comment, we have now made our data accessible and navigable in an interactive manner through two platforms, the Allen Brain Cell Atlas database and the CellxGene database, and hence readers can see the spatial locations all different cell types by exploring our data on these platforms.